# WEIGHTED-REWARD PREFERENCE OPTIMIZATION FOR IMPLICIT MODEL FUSION

**Ziyi Yang**[*]  **Fanqi Wan**[*]  **Longguang Zhong**  **Tianyuan Shi**  **Xiaojun Quan**[†]
School of Computer Science and Engineering, Sun Yat-sen University, China
`yangzy39@mail2.sysu.edu.cn, quanxj3@mail.sysu.edu.cn`

## ABSTRACT

While fusing heterogeneous open-source LLMs with varying architectures and sizes can potentially integrate the strengths of different models, existing fusion methods face significant challenges, such as vocabulary alignment and merging distribution matrices. These procedures are not only complex but also prone to introducing noise and errors. In this paper, we propose an implicit fusion method, Weighted-Reward Preference Optimization (WRPO), which leverages preference optimization between the source LLMs and the target LLM to transfer their capabilities effectively. WRPO eliminates the need for vocabulary alignment and matrix fusion and can be efficiently scaled to accommodate various LLMs. To address distributional deviations between the source and target LLMs, WRPO introduces a progressive adaptation strategy that gradually shifts reliance on preferred examples from the target LLM to the source LLMs. Extensive experiments on the MT-Bench, AlpacaEval-2, and Arena-Hard benchmarks demonstrate that WRPO consistently outperforms existing knowledge fusion methods and various fine-tuning baselines. When applied to Llama-3-8B-Instruct as the target model, WRPO achieves a length-controlled win rate of 55.9% against GPT-4-Preview-1106 on AlpacaEval-2 and a win rate of 46.2% against GPT-4-0314 on Arena-Hard. Our code is available at `https://github.com/SLIT-AI/WRPO`.

## 1 INTRODUCTION

Combining the strengths of multiple large language models (LLMs) can potentially enhance the capabilities of individual models. Model ensemble techniques (Jiang et al., 2023b; Wang et al., 2024b) aggregate predictions from several models to improve overall performance and robustness over a single model. However, this approach requires substantial computational resources, as all models must remain active during inference. The Mixture of Experts (MoE) (Komatsuzaki et al., 2023; Feng et al., 2024; Sukhbaatar et al., 2024) leverages sparse expert networks to boost capacity by activating only a subset of parameters. Despite reduced activation, MoEs still incur significant memory overhead, as all parameters must be maintained. Model merging (Wortsman et al., 2022; Matena & Raffel, 2022; Yadav et al., 2023), which combines independently trained instances of the same model through arithmetic operations, allows a single model to be maintained during inference. While more efficient, this method is restricted to models with identical architectures and sizes.

Another approach is to fuse these LLMs into a target model through multi-teacher knowledge distillation (Wan et al., 2024a;b; Shi et al., 2024). Unlike traditional knowledge distillation (Gou et al., 2021), which usually leverages diverse sources (e.g., logits, features, and relations) of knowledge from teacher models, this method relies exclusively on the probabilistic distribution matrices generated by source LLMs to transfer knowledge to the target model. We refer to this method as *explicit model fusion* (EMF) because it involves a well-defined knowledge transfer process. While applicable to heterogeneous models with varying architectures and sizes, and without increasing memory overhead during inference, this approach presents notable challenges such as vocabulary alignment and the merging of distribution matrices from different LLMs. These issues complicate model fusion, reduce its efficiency, and may introduce noise and errors and affect the fusion results.

---

[*] Contributed equally.
[†] Corresponding author.

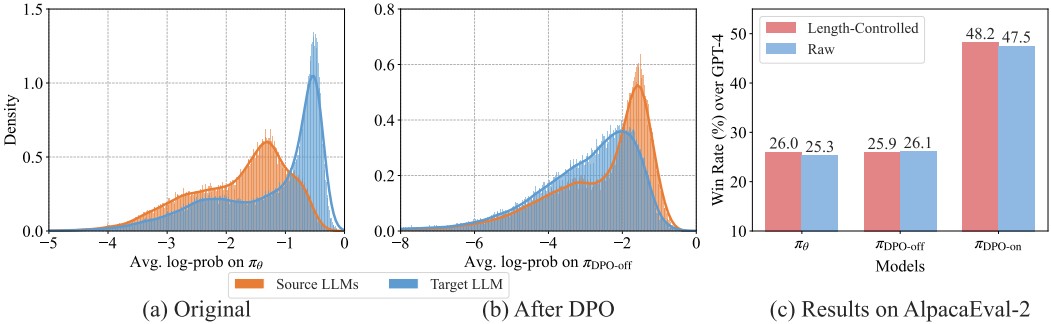

Figure 1: Distribution deviations between responses from heterogeneous source LLMs and the Llama-3-8B-Instruct target LLM before (a) and after (b) DPO fine-tuning, with the prompts from UltraFeedback (Cui et al., 2024) as input. Subfigure (c) shows the results ($\pi_{\text{DPO-off}}$) of preference optimization with this deviated preference dataset, compared to the results ($\pi_\theta$) from directly applying the target model and those ($\pi_{\text{DPO-on}}$) from DPO fine-tuning on un-deviated preference data sampled from the target model.

This work aims to enhance the capabilities of a single LLM by implicitly learning from robust open-source LLMs, a process we term *implicit model fusion* (IMF). The concept of IMF has been widely utilized to improve the performance of weaker models. For instance, a weak model can be boosted through fine-tuning with outputs from stronger LLMs (Ranaldi & Freitas, 2024; Tian et al., 2024; Kang et al., 2023). Moreover, a reward model can be trained using outputs from various LLMs (Cui et al., 2024; Zhu et al., 2024a), enabling it to learn and capture the differences in capabilities between the LLMs. Zephyr (Tunstall et al., 2023) further collects responses from multiple LLMs and ranks them with GPT-4 to obtain preference data for training the policy using DPO. One advantage of IMF over EMF (Wan et al., 2024a;b; Shi et al., 2024) is that it eliminates the need for challenging alignment of vocabularies and fusion of distributions among different LLMs. Inspired by recent alignment techniques such as Direct Preference Optimization (DPO) (Rafailov et al., 2023) and Simple Preference Optimization (SimPO) (Meng et al., 2024), we propose a novel IMF method to transfer the capabilities of source LLMs to a target LLM through preference optimization. However, directly applying preference learning to outputs from heterogeneous LLMs presents challenges. Previous works have shown that DPO is highly sensitive to distribution shifts between the policy model and the preference data (Xu et al., 2024b; Tajwar et al., 2024; Zhou et al., 2024), and training a policy model on this preference data can lead to sub-optimal performance.

To demonstrate this, we conduct a preliminary experiment on the UltraFeedback dataset (Cui et al., 2024), using Llama-3-8B-Instruct (Dubey et al., 2024) as the target model and 10 strong open-source LLMs as source models.[1] For each prompt, we first ask each source model to generate several responses and use the ArmoRM reward model (Wang et al., 2024a) to select the highest-reward response among all source LLMs as the *preferred* response, with the *dispreferred* response coming from the target LLM's completions. Figure 1(a) visualizes the average log-probability distribution of the target LLM $\pi_\theta$ for both response types, which reveals a significant deviation between the distributions of the source and target models. Although applying DPO directly on this deviated dataset marginally enhances the log-probabilities of source LLMs' responses relative to those of the target LLM, as shown in Figure 1(b), this results in sub-optimal performance compared to sampling both response types exclusively from the target LLM, as illustrated in Figure 1(c).

To address the distributional deviations during implicit model fusion, we introduce a novel approach called **W**eighted-**R**eward **P**reference **O**ptimization (WRPO). Instead of directly relying on the source LLMs to provide preferred responses, we propose a progressive adaptation strategy that begins with the target LLM providing preferred responses and gradually shifts this responsibility to source LLMs. Specifically, this progressive adaptation is implemented in two stages. First, for each prompt $x$, we construct a preference quadruple $(x, y_{w_s}, y_{w_t}, y_l)$, where $y_{w_s}$ is a preferred response generated by the source LLMs, and $y_{w_t}$ and $y_l$ are preferred and dispreferred responses, respectively, from the target LLM. Second, we gradually decrease the weight of internal rewards[2] for $y_{w_t}$ and increase the weight

---

[1]Refer to Section 4.1 for more details.

[2]We use "internal reward" to refer to the reward generated during preference optimization for preferred or dispreferred responses, in contrast to the reward provided by an external reward model.

for $y_{w_s}$ during preference optimization. This smoothing process facilitates the integration of strengths from the source models into the target model while mitigating the distributional discrepancies.

To assess the effectiveness of WRPO in implicit model fusion, we select 10 prominent open-source LLMs as the source models, with parameter sizes ranging from 9B to 236B. We chose Llama-3-8B-Instruct (Dubey et al., 2024) as the target model due to its strong performance relative to its size. Our experiments are conducted on three widely-used instruction-following benchmarks, namely, MT-Bench (Zheng et al., 2023), AlpacaEval-2 (Li et al., 2023), and Arena-Hard (Li et al., 2024). The results show that WRPO consistently outperforms existing fusion methods and various baselines. This highlights its ability to allow a model to implicitly learn from the diverse capabilities of heterogeneous LLMs while addressing distributional shifts. Notably, the fused model, Llama-3-8B-Instruct-WRPO, surpasses all source models on AlpacaEval-2 with a length-controlled win rate of 55.9%.

## 2 RELATED WORK

**Collective LLMs**   Given that LLMs are trained with various architectures and sizes on different datasets, it is reasonable to assume they possess unique capabilities and strengths. Therefore, leveraging the distinct advantages of different LLMs becomes a natural approach to developing more robust and high-capable models. Recent studies have increasingly emphasized the development of collective LLMs through the integration of diverse heterogeneous models.

LLM-Blender (Jiang et al., 2023b) presents an ensemble framework that first employs a pairwise ranking mechanism to identify the top-$K$ outputs generated by different LLMs. These selected outputs are then refined by a seq2seq model to produce enhanced results. Mixture-of-Agents (MoA) (Wang et al., 2024b) utilizes a hierarchical structure where each layer consists of multiple LLM agents. The outputs from a previous layer are concatenated and refined by each agent in the subsequent layer. However, this approach significantly increases the number of LLMs needed during inference. In addition to the sequence-level ensemble, Xu et al. (2024c) explored a token-level ensemble method that aggregates the distributions of LLMs at each decoding step through a global alignment matrix. Similarly, PackLLMs (Mavromatis et al., 2024) conducts distribution ensembling during inference utilizing sequence-level weights derived from the perplexity of each LLM on the input.

FuseLLM (Wan et al., 2024a) and FuseChat (Wan et al., 2024b) aim to fuse LLMs of various architectures and sizes into a more robust model through multi-teacher knowledge distillation. They start by aligning the vocabularies and probabilistic distributions of the source LLMs, followed by merging their distributions and continuously fine-tuning the target LLM. ProFuser (Shi et al., 2024) goes further by integrating both training mode (through cross-entropy loss) and inference mode (via model outputs), which provides a more comprehensive understanding of the capabilities of source LLMs. Although applicable to models with varying architectures and sizes, these methods face challenges such as vocabulary alignment and merging distribution matrices from different LLMs, which are complex and may also introduce noise and errors that affect the fusion results.

**Direct Preference Optimization**   Aligning LLMs with human preferences is crucial for their success. Reinforcement learning from human feedback (RLHF) (Christiano et al., 2017; Schulman et al., 2017; Ziegler et al., 2019) is a widely used approach to achieve this alignment. However, RLHF depends on complex reinforcement learning techniques such as Proximal Policy Optimization (PPO), which are challenging to implement and often unstable during training. To address these challenges, approaches such as SLiC-HF (Zhao et al., 2023) and RRHF (Yuan et al., 2023) replace reinforcement learning with a ranking loss on preference pairs to better align LLMs with human preferences, while also incorporating a regularization term based on reference responses. Similarly, DPO (Rafailov et al., 2023) directly optimizes the policy model by training the reward model on human preference data. In addition to providing more stable training, lower computational costs, and easier implementation, this approach ensures high-quality alignment with human preferences.

Subsequent research aims to address the potential limitations of DPO. For example, IPO (Azar et al., 2024) tackles the risk of overfitting by optimizing a nonlinear preference function, thus avoiding the transformation of pairwise preferences into pointwise rewards. KTO (Ethayarajh et al., 2024) is based on a new alignment objective of human-aware loss (HALO), which maximizes the utility of generated outputs directly from a binary signal indicating whether the output is desirable, rather than maximizing the likelihood of preferences. CPO (Xu et al., 2024a) and ORPO (Hong et al., 2024)

aim to eliminate the need for a reference model by streamlining the optimization process, combining supervised fine-tuning (SFT) and preference alignment into a single step. R-DPO (Park et al., 2024) introduces a length-regularization term into the DPO objective to mitigate length biases that may be exploited by DPO. Similarly, SimPO (Meng et al., 2024) revises the reward component in DPO to use the average log probability of positive or negative responses from the policy model. Another motivation for this method is that the training process aligns more closely with inference.

However, none of the above works consider the hybrid scenario where one response is generated by the policy itself while the other comes from a different LLM. This situation may introduce serious distribution shifts relative to the policy, which in turn affects the policy's optimization. The work closely related to our setup is WPO (Zhou et al., 2024), which assigns weights to off-policy preference pairs based on their likelihood under the policy model. These weights indicate the degree of deviation from the policy's distribution and mitigate the influence of preference pairs with notable deviations.

## 3 METHOD

In this section, we begin with a problem statement for implicit model fusion, followed by the preliminaries of direct preference optimization (DPO) (Rafailov et al., 2023). Finally, we provide a detailed explanation of our proposed method, Weighted-Reward Preference Optimization (WRPO).

### 3.1 PROBLEM STATEMENT

Previous works on model fusion primarily focus on transferring knowledge from various heterogeneous LLMs into a unified model via multi-teacher knowledge distillation (Wan et al., 2024a;b; Shi et al., 2024). We refer to this method as *explicit model fusion* (EMF) because it involves a well-defined knowledge transfer process. As mentioned earlier, this approach requires complex alignment of vocabularies and merging of distribution matrices across heterogeneous LLMs. In contrast, this work proposes *implicit model fusion* (IMF) to enhance the capabilities of a target LLM by implicitly learning from the outputs of robust source LLMs, thereby bypassing the difficulties of vocabulary alignment and distribution fusion. Another advantage of IMF is that the source LLMs can be either open-source or proprietary; however, for comparison with previous fusion approaches, we focus on open-source LLMs. Inspired by recent alignment techniques like DPO (Rafailov et al., 2023) and SimPO (Meng et al., 2024), we propose implementing IMF through preference optimization.

For each prompt $x_i$ in the training dataset $\mathcal{D}$, we first sample $N$ (e.g., $N$=5) responses from each of the source LLMs. Then, an external reward model is employed to identify the response with the highest reward score among all source models as *preferred*, denoted as $y_{w_s}$. Next, a *dispreferred* response can be sampled from the target LLM. However, as illustrated in Figure 1, significant deviations may exist between the distributions of the preferred and dispreferred responses, and directly applying preference optimization under these conditions could yield problematic results.

To address this issue, we propose a progressive adaptation strategy. Specifically, we sample $N$ responses from the target model and evaluate them using the reward model. The response with the highest score is labeled as another *preferred* response $y_{w_t}$, while the lowest-scoring response is regarded as the *dispreferred* response $y_l$. To tackle the challenges of distributional discrepancies and effectively utilize data from the source models, we introduce a novel optimization objective called Weighted-Reward Preference Optimization (WRPO). As shown in Figure 2, this objective introduces a fusion coefficient $\alpha$ that dynamically balances the internal reward of the preferred response $y_{w_s}$ from source models and that of $y_{w_t}$ from the target during training. This approach enables the target LLM to transition smoothly from its distribution to align with that of the source LLMs.

### 3.2 PRELIMINARIES: DIRECT PREFERENCE OPTIMIZATION

Conventional alignment methods such as reinforcement learning from human feedback (RLHF) (Christiano et al., 2017; Schulman et al., 2017; Ziegler et al., 2019) often involve complex training pipelines that are unstable and resource-intensive. In contrast, Direct Preference Optimization (DPO) (Rafailov et al., 2023) provides a more efficient alternative by fine-tuning LLMs to align with human preferences through a straightforward supervised learning objective using human-labeled preference data. DPO optimizes the policy to generate outputs that match human preferences without requiring explicit reward functions or trial-and-error updates. Specifically, DPO reformulates the

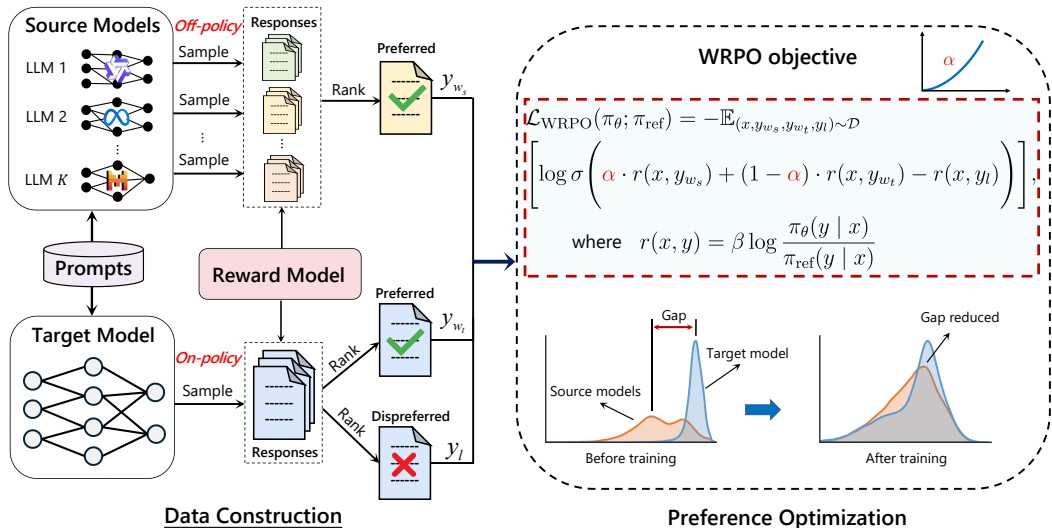

Figure 2: Overview of our proposed WRPO for implicit model fusion.

reward function to yield a closed-form solution for the optimal policy. Given the optimal policy $\pi^*$, the reparameterized form of the optimal reward function $r^*(x, y)$ is expressed as follows:

$$r^*(x, y) = \beta \log \frac{\pi^*(y \mid x)}{\pi_{\text{ref}}(y \mid x)} + \beta \log Z(x), \tag{1}$$

where $Z(x)$ is the partition function, $\pi_{\text{ref}}$ denotes the reference policy, typically a supervised fine-tuned (SFT) model, which also serves as the starting point for the policy. Given a human preference dataset $\mathcal{D} = \left\{ (x, y_w, y_l)^i \right\}_{i=1}^N$, where $y_w$ and $y_l$ represent the preferred and dispreferred completions for prompt $x$, the reparameterized reward function $r^*(x, y)$ is incorporated into the Bradley-Terry model (Bradley & Terry, 1952), which yields the probability of preference between $y_w$ and $y_l$ as:

$$p^*(y_w \succ y_l \mid x) = \sigma \left( \beta \log \frac{\pi^*(y_w \mid x)}{\pi_{\text{ref}}(y_w \mid x)} - \beta \log \frac{\pi^*(y_l \mid x)}{\pi_{\text{ref}}(y_l \mid x)} \right). \tag{2}$$

The maximum likelihood objective for a parameterized policy $\pi_\theta$ is then:

$$\mathcal{L}_{\text{DPO}}(\pi_\theta; \pi_{\text{ref}}) = -\mathbb{E}_{(x, y_w, y_l) \sim \mathcal{D}} \left[ \log \sigma \left( \beta \log \frac{\pi_\theta(y_w \mid x)}{\pi_{\text{ref}}(y_w \mid x)} - \beta \log \frac{\pi_\theta(y_l \mid x)}{\pi_{\text{ref}}(y_l \mid x)} \right) \right]. \tag{3}$$

The preference dataset for DPO training can be sampled from the reference model or sourced from publicly available data. In the latter case, a supervised fine-tuning process is typically required for the reference model to mitigate the distribution shift between the true reference distribution and the dataset used for DPO.

### 3.3 WRPO: Weighted-Reward Preference Optimization

While fine-tuning the target LLM with high-reward responses from source LLMs can alleviate the distribution issue in implicit model fusion, empirical results suggest that the distribution deviation remains, particularly when compared to preference data fully sampled from the target model. Therefore, we propose a progressive adaptation strategy with a new optimization objective called Weighted-Reward Preference Optimization (WRPO), which enables the target LLM to smoothly transition and align its distribution with that of the source LLMs.

**Derivation of the WRPO objective** The preference dataset for WRPO consists of a set of quadruples $(x, y_{w_s}, y_{w_t}, y_l)$, where $y_{w_s}$ is the highest-reward response from the source LLMs for prompt $x$, and $y_{w_t}$ and $y_l$ are the responses with the highest and lowest reward from the target LLM, respectively. Based on this setup, we define a new pair of preferred completions $\mathbf{y}_w = \{y_{w_s}, y_{w_t}\}$ and an updated preference triple $(x, \mathbf{y}_w, y_l)$. We then extend the DPO framework by introducing a weighted-reward mechanism. In particular, the Bradley-Terry (BT) model is reformulated as:

$$p(\mathbf{y}_w \succ y_l \mid x) = \sigma(r(x, \mathbf{y}_w) - r(x, y_l)), \tag{4}$$

where $r(x, \mathbf{y}_w)$ is a compound reward calculated as a weighted average of $r(x, y_{w_s})$ and $r(x, y_{w_t})$:

$$r(x, \mathbf{y}_w) = \alpha \cdot r(x, y_{w_s}) + (1 - \alpha) \cdot r(x, y_{w_t}),\qquad(5)$$

where $\alpha$ represents the fusion coefficient that dynamically balances the internal reward of the preferred response $y_{w_s}$ from source models and that of $y_{w_t}$ from the target model during training. Next, by substituting $r^*(x, y)$ from Eq. (1) into Eq. (4) and Eq. (5), we derive the WRPO training objective:

$$\mathcal{L}_{\text{WRPO}}(\pi_\theta; \pi_{\text{ref}}) = -\mathbb{E}(x, y_{w_s}, y_{w_t}, y_l) \sim \mathcal{D}$$

$$\left[\log \sigma \left(\alpha \cdot \beta \log \frac{\pi_\theta(y_{w_s} \mid x)}{\pi_{\text{ref}}(y_{w_s} \mid x)} + (1 - \alpha) \cdot \beta \log \frac{\pi_\theta(y_{w_t} \mid x)}{\pi_{\text{ref}}(y_{w_t} \mid x)} - \beta \log \frac{\pi_\theta(y_l \mid x)}{\pi_{\text{ref}}(y_l \mid x)}\right)\right], \quad(6)$$

which can be reformulated as:

$$\mathcal{L}_{\text{WRPO}}(\pi_\theta; \pi_{\text{ref}}) = -\mathbb{E}(x, y_{w_s}, y_{w_t}, y_l) \sim \mathcal{D}$$

$$\left[\log \sigma \left(\alpha \cdot \underbrace{\left(\beta \log \frac{\pi_\theta(y_{w_s} \mid x)}{\pi_{\text{ref}}(y_{w_s} \mid x)} - \beta \log \frac{\pi_\theta(y_l \mid x)}{\pi_{\text{ref}}(y_l \mid x)}\right)}_{\text{hybrid-policy internal reward margin}} + (1 - \alpha) \cdot \underbrace{\left(\beta \log \frac{\pi_\theta(y_{w_t} \mid x)}{\pi_{\text{ref}}(y_{w_t} \mid x)} - \beta \log \frac{\pi_\theta(y_l \mid x)}{\pi_{\text{ref}}(y_l \mid x)}\right)}_{\text{on-policy internal reward margin}}\right)\right].$$

$$(7)$$

The above process seeks to maximize the margin of internal rewards between preference responses, utilizing both on-policy sampling from the target model and hybrid-policy sampling from the source and target models. Initially, it emphasizes on-policy sampling and gradually transitions to hybrid-policy sampling. This process helps mitigate distributional deviations and ensures a smoother optimization process.

**Gradient analysis** We examine the gradient of WRPO to understand the impact of the weighted-reward mechanism on the training process. The gradient of loss function $\mathcal{L}_{\text{WRPO}}$ in Eq. (6) with respect to the policy model $\pi_\theta$ can be expressed as:

$$\nabla_\theta \mathcal{L}_{\text{WRPO}}(\pi_\theta; \pi_{\text{ref}}) = -\beta \mathbb{E}_{(x, y_{w_s}, y_{w_t}, y_l,) \sim \mathcal{D}} \left[\sigma \left(\underbrace{\hat{r}_\theta(x, y_l) - \alpha \cdot \hat{r}_\theta(x, y_{w_s}) - (1 - \alpha) \cdot \hat{r}_\theta(x, y_{w_t})}_{\text{higher weight when reward estimation is wrong}}\right) \cdot \right.$$

$$\left.\left(\alpha \cdot \underbrace{\nabla_\theta \log \pi_\theta(y_{w_s}|x)}_{\text{increase likelihood on } y_{w_s}} + (1 - \alpha) \cdot \underbrace{\nabla_\theta \log \pi_\theta(y_{w_t}|x)}_{\text{increase likelihood on } y_{w_t}} - \underbrace{\nabla_\theta \log \pi_\theta(y_l|x)}_{\text{decrease likelihood on } y_l}\right)\right], \quad(8)$$

where $\hat{r}_\theta(x, y) = \beta \log \frac{\pi_\theta(y|x)}{\pi_{\text{ref}}(y|x)}$ represents the internal reward function. Intuitively, the gradient flow of $\mathcal{L}_{\text{WRPO}}$ tends to increase the likelihood of preferred responses $y_{w_s}$ and $y_{w_t}$ while decreasing the likelihood of dispreferred $y_l$. The function $\sigma(.)$ represents the reward estimation error that controls the rate of increasing or decreasing the likelihood of preferred or dispreferred completions in WRPO. When the reward estimation is incorrect, WRPO will accelerate the gradient flow for updates. To further analyze the impact of $y_{w_s}$ and hyperparameter $\alpha$ on gradient update, we reformulate the $\sigma(.)$ term as $\sigma\left(\alpha \cdot (\hat{r}_\theta(x, y_l) - \hat{r}_\theta(x, y_{w_s})) + (1 - \alpha) \cdot (\hat{r}_\theta(x, y_l) - \hat{r}_\theta(x, y_{w_t}))\right)$. We can see that $\alpha$ serves as a constraint term on the gradient update for the policy model learning from $y_{w_s}$. A larger $\alpha$ means the policy will absorb more gradient information from $y_{w_s}$. At the beginning of training process, since there exists a distributional gap between $y_{w_s}$ from source models and $y_l$ from target model, we assign a relatively low $\alpha$ to the estimation term $(\hat{r}_\theta(x, y_l) - \hat{r}_\theta(x, y_{w_s}))$ and progressively increase $\alpha$ during the training process. In this way, we smoothly shift the target model $\pi_\theta$ from the distribution of $y_{w_t}$ to that of $y_{w_s}$.

Therefore, WRPO balances the contributions of diverse responses from heterogeneous LLMs and provides richer preference signals for preference optimization. Moreover, this weighted approach reduces distribution mismatches and enhances the fusion process by leveraging the strengths of both target and source models.

## 4 EXPERIMENTS

In our experiments, we use Llama-3-8B-Instruct (Dubey et al., 2024) as the target LLM. As for the source LLMs, we include ten advanced open-source models of varying architectures and sizes, as detailed in Table 1.

### 4.1 EXPERIMENTAL SETUP

**Training Dataset** Following prior work (Meng et al., 2024; Zhou et al., 2024), we chose UltraFeedback (Cui et al., 2024) to construct our training dataset. UltraFeedback includes approximately 64K prompts gathered from six established datasets that emphasize instruction-following, truthfulness, honesty, and helpfulness. However, the original dataset comprises preference data derived from old versions of LLMs, which are often less capable than our target model. Therefore, we discarded the original responses and instead used their

prompts to construct a new preference dataset $\mathcal{D}$ for our implicit model fusion, as described in Section 3.1. Specifically, for each prompt in the dataset, we sampled $N = 5$ responses from each source model using top-$p$ sampling ($p = 0.95$) with a temperature of 0.8. This approach aims to ensure that the sampled outputs capture the capabilities of the source LLMs to the greatest extent possible. ArmoRM-Llama-3-8B-v0.1 (Wang et al., 2024a) is then employed as the reward model to score and rank these responses. We selected the highest-scoring response across all source models as $y_{w_s}$, with the percentage contribution from each source LLM detailed in Table 1.

Table 1: Details of the source LLMs used in our experiments along with the percentage of the highest-scoring responses from each source LLM.

| Source LLMs | Percentage |
|---|---|
| Mistral-Large-Instruct-2407 (Jiang et al., 2023a) | 28.24% |
| Gemma-2-27B-it (Team et al., 2024) | 15.45% |
| Qwen-2-72B-Instruct (Yang et al., 2024) | 12.38% |
| Llama-3-70B-Instruct (Dubey et al., 2024) | 9.92% |
| Gemma-2-9B-it (Team et al., 2024) | 9.91% |
| InternLM-2.5-20B-Chat (Cai et al., 2024) | 7.54% |
| DeepSeek-V2-Chat (Liu et al., 2024) | 6.20% |
| DeepSeek-Coder-V2-Instruct (Zhu et al., 2024b) | 4.01% |
| Yi-1.5-34B-Chat (Young et al., 2024) | 3.86% |
| Phi-3-medium-4k-instruct (Abdin et al., 2024) | 2.49% |

**Training Details**  We conducted experiments with a batch size of 128 and a maximum length of 2048 tokens on 8x80GB NVIDIA A800 GPUs. The training was performed on a single epoch for our method. A cosine learning rate schedule with a warmup ratio of 0.1 is employed. The training process is divided into two stages. In the first stage, we applied supervised fine-tuning (SFT) on the set of $y_{w_s}$ with one-third of the dataset, with the learning rate empirically set to 7e-6. The resulting fine-tuned model, Target-SFT, is the foundation for subsequent preference optimization. In the next stage, the remaining dataset is used for preference optimization, during which $y_{w_t}$ and $y_l$ are generated from the SFT model, i.e., Target-SFT. For WRPO, we used a learning rate of 3e-7 and set $\beta = 0.01$, with the weight $\alpha$ assigned to $y_{w_s}$ linearly increasing from 0 to 0.1. Further details on hyperparameter tuning can be found in Appendix A.

**Evaluation Benchmarks**  We assess the performance of our models on three representative instruction-following benchmarks: MT-Bench (Zheng et al., 2023), AlpacaEval-2 (Li et al., 2023), and Arena-Hard (Li et al., 2024). These benchmarks are well-regarded for their comprehensive coverage of diverse tasks and their effectiveness in providing robust evaluations of the instruction-following capabilities of LLMs.

- **MT-Bench** contains 80 multi-turn dialogues with 160 questions across eight categories, including writing, roleplay, reasoning, math, coding, extraction, STEM, and humanities. Each response is evaluated by GPT-4 on a scale from 1 to 10, with the average score reported for each dialogue turn across the 80 dialogues. Different from the official setting, we follow the latest works (Wang et al., 2024c; Wan et al., 2024b) to adopt GPT-4-0125-Preview as the evaluator and baseline.

- **AlpacaEval-2** comprises 805 instructions from five different datasets and assesses models using two metrics: length-controlled (LC) win rate and raw win rate (WR) (Dubois et al., 2024). In this benchmark, GPT-4-Preview-1106 serves as both the baseline model and the evaluator for the other models.

- **Arena-Hard** is a more challenging benchmark that closely aligns with the human preference ranking from Chatbot Arena (Chiang et al., 2024), a crowd-sourced platform for evaluating LLMs. It spans 250 high-quality topic clusters including 500 well-defined technical problem-solving queries. We report the win rate against GPT-4-0314 using GPT-4-Preview-1106 as the judge model.

**Baselines**  We compare WRPO with three categories of baselines, including source&target LLMs, collective LLMs, and preference optimization methods. For **source&target LLMs**, the results are obtained from official leaderboards or our local tests if unavailable. For **collective LLMs**, we include PackLLM-Top1-PPL (Mavromatis et al., 2024), LLM-Blender-Top1 (Jiang et al., 2023b), MoA (Wang et al., 2024b), FuseLLM (Wan et al., 2024a), and FuseChat (Wan et al., 2024b). For PackLLM-Top1-PPL, we select the response from the source or target LLMs with the lowest perplexity on the test instruction. For LLM-Blender-Top1, we rank LLM outputs via pairwise comparisons and select the top response.[3] For MoA (Wang et al., 2024b), we select Mistral-Large-Instruct-2407 as the aggregator LLM to combine input responses into a single response. For FuseLLM (Wan et al., 2024a) and FuseChat (Wan et al., 2024b), limited by the complex vocabulary alignment and distribution merging process, we only include Gemma-2-27B-it, Gemma-2-9B-it, Qwen-2-72B-Instruct, Llama-3-70B-Instruct, and Yi-1.5-34B-Chat as source LLMs, with Llama-3-8B-Instruct serving as the target/pivot LLM to reimplement their methods. For a fair comparison, we select the same 5 source LLMs to implement WRPO and obtain Target-SFT-WRPO-Medium. For **preference optimization methods**, we include DPO (Rafailov et al., 2023), SimPO (Meng et al., 2024), and IPO (Azar et al., 2024). The results on AlpacaEval-2 and Arena-Hard are referenced from (Meng et al., 2024), while the results on MT-Bench are obtained by running the checkpoints released by Meng et al. (2024). In the following experimental results, these baselines are denoted as Target-DPO, Target-SimPO, and Target-IPO, respectively.

## 4.2  OVERALL RESULTS

In Table 2, we present the overall results of our WRPO method compared to various baseline methods of different categories, architectures, and scales on AlpacaEval-2, Arena-Hard, and MT-Bench benchmarks. These results offer valuable insights into WRPO's performance and efficiency as detailed below.

---

[3]Due to the *fuser* model's limited input length, we only use the *ranker* model to select the 1st-ranked output.

Table 2: Overall results of our proposed WRPO method with Llama-3-8B-Instruct as the target model, compared against various baseline categories on AlpacaEval-2, Arena-Hard, and MT-Bench. "T1" and "T2" represent the average scores for the first and second turns, respectively. **Bold** indicates the best performance in 8B models.

| Model | Size | AlpacaEval-2 (GPT-4-1106-Preview) | | Arena-Hard (GPT-4-1106-Preview) | MT-Bench (GPT-4-0125-Preview) | | |
|---|---|---|---|---|---|---|---|
| | | LC(%) | WR(%) | WR(%) | T1 | T2 | Overall |
| **Source&Target LLMs** | | | | | | | |
| Target | 8B | 26.0 | 25.3 | 20.6 | 7.41 | 7.04 | 7.23 |
| Mistral-Large-Instruct-2407 | 123B | 54.3 | 46.8 | 70.4 | 8.83 | 8.31 | 8.57 |
| Gemma-2-27B-it | 27B | 55.5 | 41.0 | 57.5 | 8.34 | 8.03 | 8.19 |
| Qwen-2-72B-Instruct | 72B | 38.1 | 29.9 | 46.9 | 8.44 | 7.84 | 8.15 |
| Llama-3-70B-Instruct | 70B | 34.4 | 33.2 | 46.6 | 8.61 | 7.77 | 8.19 |
| Gemma-2-9B-it | 9B | 51.1 | 38.1 | 40.8 | 8.27 | 7.44 | 7.86 |
| InternLM-2.5-20B-Chat | 20B | 37.4 | 45.3 | 31.2 | 8.03 | 7.23 | 7.64 |
| DeepSeek-V2-Chat | 236B | 51.4 | 51.3 | 68.3 | 8.65 | 7.96 | 8.31 |
| DeepSeek-Coder-V2-Instruct | 236B | 50.7 | 54.0 | 66.3 | 8.80 | 7.42 | 8.13 |
| Yi-1.5-34B-Chat | 34B | 37.5 | 44.5 | 42.6 | 7.99 | 7.64 | 7.81 |
| Phi-3-Medium-4K-Instruct | 14B | 29.8 | 24.2 | 33.4 | 8.63 | 7.46 | 8.04 |
| **Collective LLMs** | | | | | | | |
| PackLLM-Top1-PPL | 849B | 49.1 | 48.0 | 64.8 | 8.29 | 8.20 | 8.25 |
| LLM-Blender-Top1 | 849B | 46.2 | 44.3 | 58.2 | 8.69 | 8.06 | 8.38 |
| MoA | 849B | 61.3 | 77.2 | 83.1 | 9.04 | 8.03 | 8.54 |
| Target-FuseLLM | 8B | 36.0 | 33.8 | 32.1 | 7.53 | 7.13 | 7.33 |
| Target-FuseChat | 8B | 38.1 | 35.2 | 32.7 | 7.68 | 7.07 | 7.38 |
| **Preference Optimization Methods** | | | | | | | |
| Target-DPO | 8B | 48.2 | 47.5 | 35.2 | 7.68 | 7.23 | 7.46 |
| Target-SimPO | 8B | 53.7 | 47.5 | 36.5 | 7.73 | 7.00 | 7.38 |
| Target-IPO | 8B | 46.8 | 42.4 | 36.6 | 7.89 | 7.19 | 7.54 |
| **Our Methods** | | | | | | | |
| Target-SFT | 8B | 27.2 | 26.0 | 24.7 | 7.69 | 7.03 | 7.36 |
| Target-SFT-DPO | 8B | 50.7 | 53.1 | 40.2 | **7.98** | 7.23 | 7.61 |
| Target-SFT-WRPO-Medium | 8B | 53.5 | 53.8 | 41.6 | 7.80 | 7.03 | 7.42 |
| Target-SFT-WRPO | 8B | **55.9** | **57.6** | **46.2** | 7.95 | **7.31** | **7.63** |

**WRPO strikes a balance between effectiveness and efficiency compared to collective LLMs.** Starting with the same target LLM and involving the same source LLMs, Target-SFT-WRPO-Medium outperforms existing model fusion techniques such as FuseLLM and FuseChat by notable margins. It achieves improvements of 17.5 and 15.4 points in the length-controlled (LC) win rate on AlpacaEval-2, and 9.5 and 8.9 points in the win rate (WR) on Arena-Hard, respectively. This highlights the superior effectiveness of WRPO for implicit model fusion (IMF) compared to previous explicit model fusion (EMF) methods. Particularly, our fused model, Target-SFT-WRPO, surpasses all larger source LLMs on AlpacaEval-2, showcasing WRPO's potential to enable a target model to outperform much larger models. Furthermore, compared to collective LLM fusion architectures that are 106 times larger in scale, WRPO outperforms most of these models, only falling short of MoA on AlpacaEval-2, while incurring substantially lower computational costs. While WRPO may not exceed the absolute performance of larger ensemble systems across all evaluation metrics, its ability to achieve comparable results with far lower computational demands presents an elegant solution to the ongoing efficiency-effectiveness trade-off in language model deployment.

**WRPO consistently outperforms preference optimization baselines.** In terms of preference optimization, WRPO delivers notable improvements over prior methods. After fine-tuning $y_{w_s}$ using one-third of the data, Target-SFT performs slightly better than the target model. Following further optimization on the remaining two-thirds of the dataset, WRPO consistently outperforms all preference optimization baselines. Specifically, WRPO outperforms the best-performing preference optimization baseline on three benchmarks by 2.2, 9.6, and 0.09 points, respectively. Besides, starting from Target-SFT, WRPO achieves 5.2 points improvement over DPO in the length-controlled win rate on AlpacaEval-2, and a 6.0 points increase in win rate on Arena-Hard. Compared to these approaches which utilize responses exclusively from the target LLM, the proposed WRPO method effectively incorporates responses sampled from various source LLMs for preference optimization, thus facilitating the integration of diverse knowledge and capabilities through implicit model fusion.

## 4.3 ADAPTABILITY OF WRPO TO VARIED OBJECTIVES AND SOURCE LLM SCALING

In this section, we examine how WRPO adapts to diverse preference optimization objectives and scales with varying numbers of source LLMs, demonstrating its flexibility in both dimensions.

**Adaptation to different preference optimization objectives** Beyond DPO, we also investigate integrating our WRPO mechanism with alternative preference optimization objectives, utilizing the same SFT target model as the above experiments for DPO. Specifically, we experiment with IPO, which employs a similar internal reward formulation to DPO but optimizes a nonlinear objective, as well as SimPO, which defines its reward function based on the average log-likelihood of a response, thereby eliminating the need for a reference model. Detailed descriptions of the training objectives and the hyperparameter search ranges for these methods are provided in Appendix A.

Table 3: Results of WRPO combined with different preference optimization objectives.

| Method | AlpacaEval-2 | | MT-Bench |
|---|---|---|---|
| | LC(%) | WR(%) | Overall |
| SimPO | 53.9 | 49.9 | 7.39 |
| IPO | 51.1 | 52.4 | 7.67 |
| WRPO$_{SimPO}$ | 55.8 | 51.8 | 7.42 |
| WRPO$_{IPO}$ | 53.3 | 57.7 | 7.72 |

We refer to the methods combining WRPO with SimPO and IPO as WRPO$_{SimPO}$ and WRPO$_{IPO}$, respectively. The performance of these methods on AlpacaEval-2 and MT-Bench is summarized in Table 3. We note that combining WRPO with IPO and SimPO consistently improves their performance, highlighting our WRPO's generalizability and efficacy in integrating preference signals from heterogeneous LLMs into the target LLM across various preference optimization objectives.

**Scaling with different numbers of source LLMs** We conduct experiments with varying numbers of source LLMs to implement the WRPO framework. For the five source LLMs configuration, we select Gemma-2-27B-it, Gemma-2-9B-it, Qwen-2-72B-Instruct, Llama-3-70B-Instruct, and Yi-1.5-34B-Chat, aligning our setup with the comparisons made in FuseLLM and FuseChat. Moreover, we utilize two subsets of these five source LLMs for experiments involving fewer source LLMs. One subset includes a single LLM, Gemma-2-27B-it, while the other consists of two LLMs: Gemma-2-27B-it and Qwen-2-72B-Instruct. The results in Table 4 show that the performance of WRPO exhibits an overall upward trend as the number of source LLMs

Table 4: Results of our WRPO implemented with varying numbers of source LLMs on AlpacaEval-2 and MT-Bench.

| Num | AlpacaEval-2 | | MT-Bench |
|---|---|---|---|
| | LC(%) | WR(%) | Overall |
| 1 | 48.9 | 50.3 | 7.29 |
| 2 | 52.3 | 50.4 | 7.54 |
| 5 | 53.5 | 53.8 | 7.42 |
| 10 | 55.9 | 58.0 | 7.63 |

increases on AlpacaEval-2 and MT-Bench. This trend demonstrates the potential effectiveness of scaling up the number of source LLMs to enhance our method.

## 4.4 ANALYSIS OF THE WEIGHTED-REWARD MECHANISM IN WRPO

In this section, we conduct an in-depth analysis of the weighted-reward mechanism in our implicit model fusion framework, focusing on three distinctive views.

**Balancing internal reward dynamics** Figure 3 demonstrates the evolution of internal reward dynamics during preference optimization in the Target-SFT model across various preference pairs, with consistent learning rate and $\beta$ parameters. The internal reward margin, as defined in Eq. (7), comprises an on-policy reward margin $r(x, y_{w_t}) - r(x, y_l)$ weighted by $1 - \alpha$, and a hybrid-policy reward margin $r(x, y_{w_s}) - r(x, y_l)$ weighted by $\alpha$. Figure 3(a) presents the analysis of solely utilizing the on-policy reward margin ($\alpha = 0$). The observed reward margin approximates 0.2, indicating a relatively conservative optimization approach. This modest margin growth can be attributed to the model's limited exploration capability due to its exclusive reliance on on-policy samples. In contrast, Figure 3(b) illustrates the effect of employing only the hybrid-policy reward margin ($\alpha = 1$). This configuration exhibits more aggressive optimization behavior, yielding reward margins exceeding 1.0. While this suggests enhanced discriminative capability between positive and negative samples, the substantial distribution shift inherent in the hybrid setting may compromise training stability and ultimately yield suboptimal results. Figure 3(c) showcases our proposed weighted-reward mechanism, which synthesizes both on-policy and hybrid-policy reward margins through dynamic weighting. This approach achieves an optimal balance between the aforementioned extremes, generating moderate reward margins of approximately 0.5 and facilitating smooth margin transitions throughout the training process. The harmonious integration of on-policy and hybrid-policy components, as evidenced by the balanced optimization process, appears to be instrumental in the superior performance of our weighted-reward mechanism.

**Effectiveness of weighted-reward mechanism** Figure 4 illustrates the ablation studies on the effectiveness of incorporating preferred responses from both source and target LLMs. We conduct these studies on two configurations: the baseline target model (Target) and its fine-tuned version (Target-SFT) to ensure a comprehensive evaluation. The analysis involves systematically removing either the source model's chosen response $y_{w_s}$ or the target model's chosen response $y_{w_t}$ from the optimization objective in Eq. (6) by setting $\alpha = 0$ or $\alpha = 1$, respectively. In the Target setting, the removal of $y_{w_t}$ leads to a substantial decline of 25.8 points in the length-controlled win rate, indicating that the distribution shift between $y_{w_s}$ and $y_l$ creates challenges in directly utilizing source model responses for preference optimization. Moreover, this finding emphasizes the crucial role of $y_{w_t}$ in bridging this distribution gap. In the Target-SFT setting, although SFT helps mitigate the performance deterioration caused by removing $y_{w_t}$, its performance still lags behind our

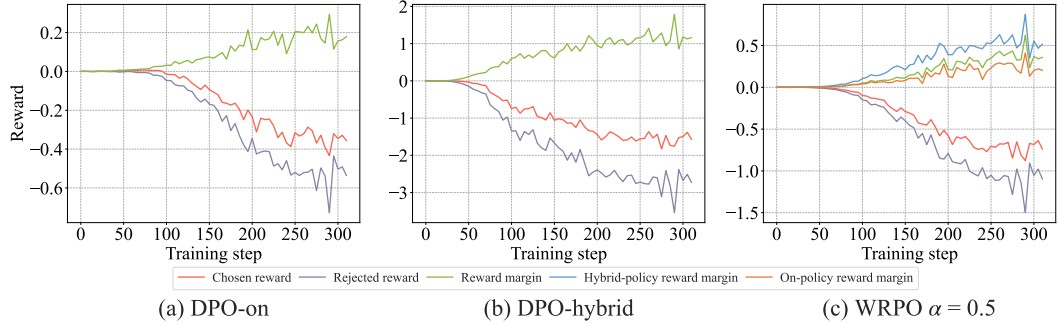

(a) DPO-on        (b) DPO-hybrid        (c) WRPO $\alpha = 0.5$

Figure 3: Internal reward dynamics on Target-SFT model under different preference optimization setups. (a) DPO-on: DPO training on on-policy preference pairs $(x, y_{w_t}, y_l)$. (b) DPO-hybrid: DPO training on hybrid-policy preference pairs $(x, y_{w_s}, y_l)$. (c) WRPO $\alpha = 0.5$: WRPO training with $\alpha$ increasing from 0 to 0.5.

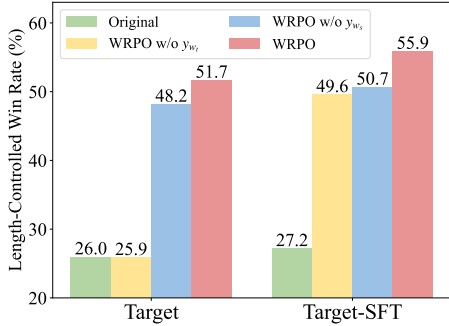

Figure 4: Results of ablation studies for our WRPO method on AlpacaEval-2, utilizing the length-controlled win rate metric.

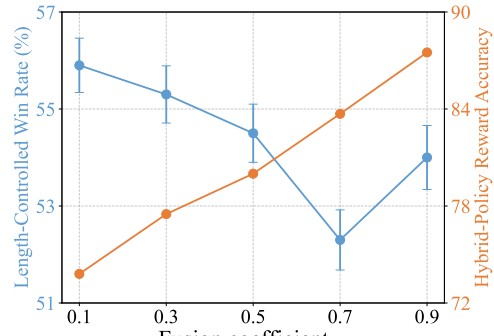

Figure 5: AlpacaEval-2 length-controlled win rate and hybrid-policy reward accuracy under different fusion coefficient $\alpha$ settings.

WRPO by 6.3 points, which combines both $y_{w_s}$ and $y_{w_t}$. On the other hand, removing $y_{w_s}$ reduces WRPO to DPO based solely on self-sampled on-policy data. Notably, the exclusion of source model responses leads to performance declines of 3.5 points and 5.2 points in the Target and Target-SFT settings, respectively, highlighting the important role of $y_{w_s}$ in providing valuable preference signals through the weighted-reward mechanism.

**Influence of fusion coefficient** We evaluate the impact of varying the fusion coefficient $\alpha$ in the weighted-reward mechanism, with $\alpha \in [0.1, 0.3, 0.5, 0.7, 0.9]$, by recording the length-controlled (LC) win rate on AlpacaEval-2 and the hybrid-policy reward accuracy on a held-out set of the UltraFeedback dataset. Hybrid-policy reward accuracy is defined as the percentage of instances where the internal reward $r(x, y_{w_s})$ from source LLMs surpasses $r(x, y_l)$ from the target LLM. As shown in Figure 5, hybrid-policy reward accuracy improves as $\alpha$ increases, indicating that progressively increasing $\alpha$ over a wide range leads to higher hybrid-policy reward accuracy. However, the LC win rate on AlpacaEval-2 shows an initial decline followed by an improvement. This suggests that although increasing $\alpha$ may provide richer preference signals and achieve higher hybrid-policy reward accuracy on the UltraFeedback held-out set, it does not always correlate with real-world benchmark performance. Nonetheless, WRPO consistently outperforms the DPO baseline (50.7) across all $\alpha$ settings.

## 5 CONCLUSION

In this work, we introduce Weighted-Reward Preference Optimization (WRPO) for the implicit model fusion of heterogeneous open-source LLMs with diverse architectures and sizes, aiming to create a more capable and robust target LLM. To address distributional deviations between source and target LLMs, WRPO utilizes a progressive adaptation strategy that gradually shifts reliance on preferred responses from the target LLM to the source LLMs. Extensive experiments on three public benchmarks demonstrate that WRPO consistently outperforms existing knowledge fusion methods and various fine-tuning baselines.

This study concludes with three notable findings. First, implicit model fusion presents a promising approach to enhancing the capabilities of LLMs by eliminating the need for vocabulary alignment and distribution merging. Second, the fusion of LLMs can be redefined as a preference optimization task, distinguishing it from conventional methods such as knowledge distillation and fine-tuning. Finally, our WRPO effectively addresses challenges related to hybrid-policy sampling, enabling efficient scaling to accommodate various LLMs.

## ACKNOWLEDGEMENTS

This work was supported by the National Natural Science Foundation of China (No. 62176270) and the Guangdong Basic and Applied Basic Research Foundation (No. 2023A1515012832).

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

# A    HYPERPARAMETER TUNING

Table 5: Various preference optimization objectives and hyperparameter search range.

| Method | Objective | Hyperparameter |
|---|---|---|
| DPO (Rafailov et al., 2023) | $-\log\sigma\left(\beta\log\frac{\pi_\theta(y_w|x)}{\pi_{\text{ref}}(y_w|x)} - \beta\log\frac{\pi_\theta(y_l|x)}{\pi_{\text{ref}}(y_l|x)}\right)$ | $\beta \in [0.01, 0.05, 0.1]$ |
| SimPO (Meng et al., 2024) | $-\log\sigma\left(\frac{\beta}{|y_w|}\log\pi_\theta(y_w|x) - \frac{\beta}{|y_l|}\log\pi_\theta(y_l|x) - \gamma\right)$ | $\beta \in [5.0, 10.0]$ $\gamma \in [0, 1.0, 2.0]$ |
| IPO (Azar et al., 2024) | $\left(\log\frac{\pi_\theta(y_w|x)}{\pi_{\text{ref}}(y_w|x)} - \log\frac{\pi_\theta(y_l|x)}{\pi_{\text{ref}}(y_l|x)} - \frac{1}{2\tau}\right)^2$ | $\tau \in [0.01, 0.1, 1.0]$ |
| WRPO$_{\text{DPO}}$ | $-\log\sigma\left(\alpha\cdot\beta\log\frac{\pi_\theta(y_{w_s}|x)}{\pi_{\text{ref}}(y_{w_s}|x)} + (1-\alpha)\cdot\beta\log\frac{\pi_\theta(y_{w_t}|x)}{\pi_{\text{ref}}(y_{w_t}|x)} - \beta\log\frac{\pi_\theta(y_l|x)}{\pi_{\text{ref}}(y_l|x)}\right)$ | $\beta = 0.01$ $\alpha \in [0.1, 0.3, 0.5, 0.7, 0.9]$ |
| WRPO$_{\text{SimPO}}$ | $-\log\sigma\left(\alpha\cdot\frac{\beta}{|y_{w_s}|}\log\pi_\theta(y_{w_s}|x) + (1-\alpha)\cdot\frac{\beta}{|y_{w_t}|}\log\pi_\theta(y_{w_t}|x) - \frac{\beta}{|y_l|}\log\pi_\theta(y_l|x) - \gamma\right)$ | $\beta = 10.0, \gamma = 0$ $\alpha \in [0.1, 0.3, 0.5]$ |
| WRPO$_{\text{IPO}}$ | $\left(\alpha\cdot\log\frac{\pi_\theta(y_{w_s}|x)}{\pi_{\text{ref}}(y_{w_s}|x)} + (1-\alpha)\cdot\log\frac{\pi_\theta(y_{w_t}|x)}{\pi_{\text{ref}}(y_{w_t}|x)} - \log\frac{\pi_\theta(y_l|x)}{\pi_{\text{ref}}(y_l|x)} - \frac{1}{2\tau}\right)^2$ | $\tau \in [0.01, 0.1]$ $\alpha \in [0.1, 0.3, 0.5]$ |

Prior works such as SimPO (Meng et al., 2024) suggest that hyperparameter tuning is crucial for achieving optimal performance of preference optimization methods. To avoid getting suboptimal baseline results, we followed the recommendation by Meng et al. (2024) to apply hyperparameter tuning for all preference optimization methods, including DPO (Rafailov et al., 2023), SimPO (Meng et al., 2024), and IPO (Azar et al., 2024). Specifically, we individually search the learning rates in the range of [3e-7, 5e-7, 6e-7, 1e-6] for each preference optimization method. The specific training objectives and hyperparameter search ranges for these preference optimization baselines, along with our method, are outlined in Table 5. We used a batch size of 128 and trained these methods for a single epoch. The best hyperparameter values under the Target-SFT setting are summarized

Table 6: Hyperparameter settings for preference optimization methods using Target-SFT as the policy model. "LR" denotes the learning rate.

| Method | $\beta$ | $\gamma$ | $\alpha$ | LR |
|---|---|---|---|---|
| DPO | 0.01 | - | - | 3e-7 |
| SimPO | 10 | 1.0 | - | 6e-7 |
| IPO | - | - | 0.01 | 1e-6 |
| WRPO$_{\text{DPO}}$ | 0.01 | - | 0.1 | 3e-7 |
| WRPO$_{\text{SimPO}}$ | 10 | 0 | 0.5 | 6e-7 |
| WRPO$_{\text{IPO}}$ | - | 0.01 | 0.1 | 1e-6 |

in Table 6. Besides, a learning rate of 7e-6 was used with a single epoch for supervised fine-tuning (SFT). For the model fusion methods, including FuseLLM (Wan et al., 2024a) and FuseChat (Wan et al., 2024b), we used a learning of 7e-6 and conducted training over three epochs, with the parameter $\lambda$ empirically set to 0.9.

# B    EVALUATION ON ADDITIONAL BENCHMARKS

To further investigate the impact of WRPO on downstream tasks, we evaluate the models we trained using six tasks from the Huggingface Open LLM Leaderboard (Beeching et al., 2023). These tasks include:

**AI2 Reasoning Challenge (ARC)** (Clark et al., 2018): A collection of grade-school science questions in a 25-shot setting.

**HellaSwag** (Zellers et al., 2019): A commonsense inference task in a 10-shot setting.

**MMLU** (Hendrycks et al., 2021): A set of 57 diverse tasks spanning high-school and college subjects, social sciences, STEM, and others, in a 5-shot setting.

**TruthfulQA** (Lin et al., 2022): A set of measuring how language models mimic human falsehoods with 6-shot setting[4].

**Winogrande** (Levesque et al., 2012): A set of adversarial and difficult Winograd benchmarks for commonsense reasoning in a 5-shot setting.

**GSM8K** (Cobbe et al., 2021): A set of grade-school math word questions evaluates mathematical reasoning capabilities in a 5-shot setting.

We followed the established evaluation pipelines by using the *lm-evaluation-harness* tool[5] for our evaluation. The results are presented in Table 7, from which we draw several key observations. Firstly, after undergoing SFT with one-third of the data entries, Target-SFT shows a significant performance decline compared to Target, particularly on ARC and GSM8K, likely due to catastrophic forgetting during training. Next, all preference optimization methods display varying performance drops on MMLU and GSM8K, which may stem from the UltraFeedback

---

[4]Although TruthfulQA is traditionally regarded as 0-shot, it is technically a 6-shot task because each example is associated with 6 Q&A pairs.

[5]We used an updated version of v0.4.3 at `https://github.com/EleutherAI/lm-evaluation-harness/tree/v0.4.3` for more accurate evaluation.

Table 7: Results of evaluations on Huggingface Open LLM Leaderboard. "Target" denotes Llama-3-8B-Instruct.

| Model | ARC | HellaSwag | MMLU | TruthfulQA | Winogrande | GSM8K | Avg. |
|---|---|---|---|---|---|---|---|
| Target | 61.43 | 78.48 | **65.71** | 51.64 | 75.61 | **75.21** | 68.01 |
| Target-SFT | 51.19 | 79.83 | 64.56 | 45.93 | 76.87 | 62.77 | 63.53 |
| Target-SFT-DPO | 60.67 | 81.7 | 64.98 | 50.3 | 76.95 | 68.76 | 67.23 |
| Target-SFT-SimPO | 61.77 | 82.23 | 65.13 | 54.76 | 78.45 | 69.6 | 68.66 |
| Target-SFT-IPO | 60.58 | 81.68 | 65.5 | 53.93 | 77.9 | 69.67 | 68.21 |
| Target-SFT-WRPO | **62.63** | **82.38** | 64.91 | 54.72 | 78.53 | 71.57 | **69.12** |
| Target-SFT-WRPO$_{SimPO}$ | 61.69 | 81.95 | 65.08 | **57.11** | **78.69** | 68.69 | 68.87 |
| Target-SFT-WRPO$_{IPO}$ | 59.98 | 81.53 | 65.35 | 53.48 | 78.14 | 69.83 | 68.05 |

dataset's focus on alignment over general knowledge and mathematics. In contrast, these preference optimization methods consistently improve performance on HellaSwag and Winogrande, suggesting the presence of relevant prompts for commonsense inference in UltraFeedback. Similarly, all preference optimization methods show consistent gains on TruthfulQA, except for Target-SFT-DPO. Lastly, the performance of ARC demonstrates only minor improvements or declines across all methods. In summary, while not explicitly designed for these tasks, our fused model, Target-SFT-WRPO, surpasses the initial Target model while preserving general knowledge and mathematical abilities with minimal decline. This illustrates the generalization potential of our WRPO method.

## C  TRAINING COST ANALYSIS

Increasing the number of source LLMs does not affect the time complexity of our method during training. First, the interaction with source LLMs occurs exclusively during the data collection phase before training, where we conduct offline sampling from the source LLMs and utilize ArmoRM as a reward model to evaluate the responses and select one response with the highest reward score for each prompt. This step constitutes a fixed, one-time computational cost that is independent of the training process. Importantly, the source LLMs do not participate in the actual training phase. Therefore, the inclusion of additional source LLMs does not introduce additional computational costs during WRPO training. Furthermore, our comparative analysis in Table 8 shows that WRPO maintains consistent computational efficiency across different numbers of source LLMs. Notably, WRPO incurs only a modest overhead of approximately 16% in training time compared to DPO (which does not involve source LLMs) on 8×A800 GPUs, regardless of the number of source LLMs involved.

Table 8: Runtime comparisons for DPO and WRPO across different numbers of source LLMs.

| Num. | Runtime of DPO (min) | Runtime of WRPO (min) | Increase (%) |
|---|---|---|---|
| 1 | 183 | 212 | 15.88% |
| 2 | 185 | 215 | 16.22% |
| 5 | 186 | 216 | 16.13% |
| 10 | 185 | 215 | 16.22% |

## D  DIFFERENT COMBINATIONS OF SOURCE LLMS

To explore the impact of different source model combinations, we conducted additional experiments using the AlpacaEval-2 benchmark. Specifically, we examined the influence of the response quality from source LLMs by comparing responses with different reward rankings. The experimental results in Table 9 indicate that responses from top-ranked source models consistently outperform those from second-ranked models. This reinforces the importance of selecting high-quality responses to achieve optimal performance. In addition, we investigated the impact of model composition by dividing our ten source models into two balanced groups, each comprising five models with strong performance characteristics. The first group includes Gemma-2-27B-it, Gemma-2-9B-it, Qwen-2-72B-Instruct, Llama-3-70B-Instruct,

Table 9: Results of our WRPO implemented with varying combinations of source LLMs on AlpacaEval-2.

| Method | AlpacaEval-2 | | |
|---|---|---|---|
| | LC(%) | WR(%) | Length |
| Rank1 | 55.9 | 57.6 | 2159 |
| Rank2 | 53.7 | 55.4 | 2143 |
| Group1 | 53.5 | 53.8 | 2098 |
| Group2 | 53.7 | 60.7 | 2440 |

and Yi-1.5-34B-Chat. The second group comprises Mistral-Large-Instruct-2407, InternLM-2.5-20B-Chat, DeepSeek-V2-Chat, DeepSeek-Coder-V2-Instruct, and Phi-3-Medium-4K-Instruct. The experimental results in Table 9 show that various combinations of source models achieve comparable length-controlled (LC) win rate. These findings demonstrate the robust performance of WRPO across a range of source model configurations.

## E    TUNING STRATEGIES FOR FUSION COEFFICIENT

In WPRO, we implement a dynamic adjustment mechanism for the fusion coefficient $\alpha$ to facilitate a gradual transition of the target model's distribution toward that of the source models. In practice, the fusion coefficient $\alpha$ is initialized at 0.0 and increases linearly throughout the training process until it reaches a pre-determined target value (Clark et al., 2019). To determine the optimal target value, we employ a simple greedy search over the range [0.1, 0.3, 0.5, 0.7, 0.9]. This dynamic adjustment strategy effectively balances the contributions from both source and target models while addressing potential distribution discrepancies, making it suitable for various tasks and eliminating the need for complex parameter configurations or exhaustive optimization procedures. Moreover, we conducted ablation experiments comparing static and dynamic tuning strategies. In the static strategy, $\alpha$ remains fixed at a target value throughout training, while in the dynamic strategy, $\alpha$ linearly increases from 0 to the target value. The experimental results in Figure 6 show that the dynamic tuning strategy generally outperforms the static strategy, except for setting $\alpha = 0.7$, further demonstrating the effectiveness of the dynamic tuning approach.

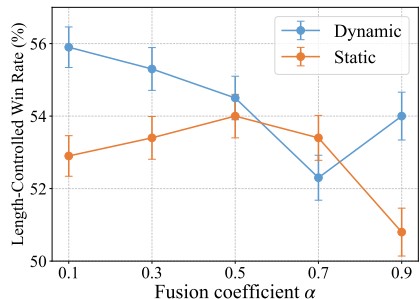

Figure 6: Comparisons of dynamic and static tuning strategies for the fusion coefficient on AlpacaEval-2, utilizing the length-controlled win rate metric.

## F    INCLUDING DISPREFERRED RESPONSES FROM SOURCE MODELS

We conducted additional experiments to investigate the impact of incorporating extra dispreferred responses from the source models. Specifically, we use an extension of the WRPO loss in Eq. (9), where $y_{l_t}$ denotes the dispreferred response from the target model, and $y_{l_s}$ denotes the dispreferred response from the same source model corresponding to the preferred response $y_{w_s}$.

$$
\mathcal{L}_{\mathrm{WRPO}_{w/y_{l_s}}} (\pi_\theta; \pi_{\mathrm{ref}}) = -\mathbb{E}(x, y_{w_s}, y_{w_t}, y_{l_s}, y_{l_t}) \sim \mathcal{D} \left[ \log \sigma \left( \alpha \cdot \beta \log \frac{\pi_\theta(y_{w_s} \mid x)}{\pi_{\mathrm{ref}}(y_{w_s} \mid x)} \right. \right.
$$

$$
\left. \left. + (1-\alpha) \cdot \beta \log \frac{\pi_\theta(y_{w_t} \mid x)}{\pi_{\mathrm{ref}}(y_{w_t} \mid x)} - \alpha \cdot \beta \log \frac{\pi_\theta(y_{l_s} \mid x)}{\pi_{\mathrm{ref}}(y_{l_s} \mid x)} - (1-\alpha) \cdot \beta \log \frac{\pi_\theta(y_{l_t} \mid x)}{\pi_{\mathrm{ref}}(y_{l_t} \mid x)} \right) \right]. \quad (9)
$$

First, we perform a comparative analysis of the reward scores across four categories of responses. The average reward scores for $y_{w_s}$, $y_{w_t}$, $y_{l_s}$, and $y_{l_t}$ are 0.180, 0.152, 0.158, and 0.132, respectively. We observe that the source model's dispreferred responses $y_{l_s}$ have higher average scores than the target model's preferred responses $y_{w_t}$. This finding indicates that incorporating dispreferred responses from source models into the training objective could potentially lead to an undesirable reduction in the probability of higher-scoring responses. Such results would contradict our optimization objectives and potentially compro-

Table 10: Results of WRPO combined with additional dispreferred responses from source models.

| Method | AlpacaEval-2 | | MT-Bench |
|---|---|---|---|
| | LC(%) | WR(%) | Overall |
| WRPO | 55.9 | 57.6 | 7.63 |
| WRPO$_{w/y_{l_s}}$ | 54.0 | 56.0 | 7.52 |

mise the overall training effectiveness. Furthermore, the results in Table 10 show that the inclusion of rejected responses from the source model leads to a decrease in performance on AlpacaEval-2 and MT-Bench. Moreover, this approach increases computational costs due to the need for extra forward passes during training.

## G    DETAILS OF OPEN-SOURCE MODELS AND THE DATASET

The selection of source models and the dataset is primarily determined by specific objectives. When a target model exhibits limitations in particular domains, domain-specific source models and datasets can be strategically used to enhance its capabilities. In our study, we focus on instruction-following tasks to align with prior preference optimization research. Therefore, we selected ten prominent open-source LLMs with parameter sizes ranging from 9B to 236B, all of which exhibit strong performance on relevant benchmarks. Moreover, we chose one of the most popular instruction-following datasets, the UltraFeedback (Cui et al., 2024), as our training dataset. In Table 11, we provide the Huggingface repository names of the target LLM, source LLMs, reward model, and preference optimization baseline checkpoints used in our experiments. For the UltraFeedback (Cui et al., 2024) dataset, we select the same prompts as provided by Meng et al. (2024) in princeton-nlp/llama3-ultrafeedback-armorm for fair comparison to baselines.

Table 11: Details of open-source models in our experiments. "Target" denotes Llama-3-8B-Instruct.

| Model | Huggingface ID |
|---|---|
| Target | meta-llama/Meta-Llama-3-8B-Instruct |
| Mistral-Large-Instruct-2407 | Mistral-Large-Instruct-2407 |
| Gemma-2-27B-it | google/gemma-2-27b-it |
| Qwen-2-72B-Instruct | Qwen/Qwen2-72B-Instruct |
| Llama-3-70B-Instruct | meta-llama/Meta-Llama-3-70B-Instruct |
| Gemma-2-9B-it | google/gemma-2-9b-it |
| InternLM-2.5-20B-Chat | internlm/internlm2_5-20b-chat |
| DeepSeek-V2-Chat | deepseek-ai/DeepSeek-V2-Chat-0628 |
| DeepSeek-Coder-V2-Instruct | deepseek-ai/DeepSeek-Coder-V2-Instruct-0724 |
| Yi-1.5-34B-Chat | 01-ai/Yi-1.5-34B-Chat |
| Phi-3-medium-4k-instruct | microsoft/Phi-3-medium-4k-instruct |
| ArmoRM-Llama-3-8B-v0.1 | RLHFlow/ArmoRM-Llama3-8B-v0.1 |
| Target-DPO | princeton-nlp/Llama-3-Instruct-8B-DPO-v0.2 |
| Target-SimPO | princeton-nlp/Llama-3-Instruct-8B-SimPO-v0.2 |
| Target-IPO | princeton-nlp/Llama-3-Instruct-8B-IPO-v0.2 |

## H  LIMITAIONS AND FUTURE WORK

First, the WRPO training objective currently incorporates only the highest-scoring response from source models as the preferred output for each prompt. This selective approach may overlook other valuable responses, potentially underutilizing the full range of capabilities offered by the source models. Future work could explore more inclusive methods that incorporate multiple responses from source models into the training objective. Second, while WRPO demonstrates strong empirical performance, it relies heavily on existing preference optimization frameworks. A more rigorous theoretical analysis is needed to provide deeper insights into the internal fusion dynamics of WRPO and to further strengthen its theoretical foundation. Finally, while WRPO significantly improves performance on instruction-following tasks, it may not perform as well on other tasks, such as MMLU. This limitation can largely be attributed to the narrow domain coverage of the training dataset. Future studies could address this by incorporating more diverse datasets from a wider range of domains.

## I  CASE STUDY

Finally, we evaluate the performance of our WRPO method using a sample instruction from Arena-Hard (Li et al., 2024), which features a math puzzle centered on Situation Puzzle Challenges. Using Llama-3-8B-Instruct as the target model, we compare the performance of Target-SFT, Target-SFT-DPO, and Target-SFT-WRPO against the baseline model GPT-4-0314. As shown in table 12, Target-SFT misinterprets the puzzle, providing an incorrect answer of 3 dots as the maximum. In contrast, Target-SFT-DPO approaches the problem through a graph theory, arriving at a solution of 5 dots. Although this answer is logically derived from graph theory, it remains incorrect. Notably, Target-SFT-WRPO demonstrates significant improvement over the other models, surpassing the GPT-4-0314 baseline by correctly solving the puzzle and offering a comprehensive explanation. This response not only accurately addresses the puzzle but also discusses practical limitations related to page size and provides effective strategies for solving similar puzzles in real-world contexts. The combination of factually sound reasoning and a thorough, user-friendly explanation underscores the superiority of the WRPO approach.

Table 12: Case study of a math puzzle on Situation Puzzle Challenges from Arena-Hard. Our Target-SFT-WRPO model delivers a mathematically sound and user-oriented response, highlighting the efficacy of WRPO in enhancing model performance in challenging problem-solving scenarios.

---

**Topic: Situation Puzzle Challenges**

**Instruction:** Help me find out how to solve a math puzzle. It's: "How many dots you can joint down on a page while drawing lines between them, without three of them ever forming a straight line?"

**GPT-4-0314 Reference Answer:** This math puzzle is related to the concept of "no three points being collinear." In this case, you want to find the maximum number of points you can place on a page such that no three of them are in a straight line. The answer to this puzzle is that you can place an infinite number of points without having three of them form a straight line. One way to achieve this is by placing the points in a non-linear pattern, such as arranging them in a spiral or a circle. However, if you are looking for a more specific answer, you can consider placing the points in a triangular grid pattern. In this arrangement, you can place as many points as you want without having three of them form a straight line. Each new point added will be connected to the existing points without creating a straight line with any other two points.

**Target-SFT Answer:** This is a classic puzzle known as the "Three Dots Puzzle" or "No Straight Lines Puzzle." The answer is 3. Here's why: If you draw lines between 3 dots, you can't avoid creating at least one straight line. If you draw lines between 4 dots, you can create a triangle with two straight sides, which means three dots are forming a straight line. If you draw lines between 5 dots, you can create a pentagon with three straight sides, which means three dots are forming a straight line. Therefore, the maximum number of dots you can connect without three of them forming a straight line is 3.

**Target-SFT-DPO Answer:** This puzzle is a classic example of a graph theory problem, specifically a planar graph challenge. Here's a breakdown and solution: Understanding the Problem:
Graph: Think of each dot as a node, and the lines connecting them as edges.
Planarity: The key constraint is that no three nodes (dots) can be connected by straight lines, forming a triangle. This ensures no three edges intersect at a single point.
Goal: Find the maximum number of dots you can connect without violating this planarity condition.
Solution: The maximum number of dots you can connect while maintaining planarity is 5. Why 5?
Proof: Imagine arranging the dots in a pentagon (a five-sided shape). Each edge of the pentagon connects two dots, and no three dots form a straight line within the pentagon.
Generalization: For any number of dots greater than 5, it's impossible to draw a planar graph without three dots forming a straight line. This is because as you add more dots, the likelihood of forming triangles increases, eventually leading to intersections.
Key Takeaway: For any planar graph, the maximum number of nodes (dots) without three forming a straight line is 5. This fundamental limit arises from the inherent geometry of planar graphs.
I hope this explanation helps you solve your math puzzle!

**Target-SFT-WRPO Answer:** Unraveling the Puzzle: This problem is a classic example of "Non-Collinear Point Arrangement" with a twist. Let's break it down step by step:
Key Insights:
1. No Straight Triangles: The core constraint is that no three dots (points) should ever form a straight line.
2. Geometric Considerations:
- A straight line requires three points in a specific arrangement (collinear).
- We need to avoid this configuration.
Strategies for Solving:
1. The Fundamental Approach:
Inductive Reasoning: Start with two dots. Adding a third dot that doesn't create a straight line is always possible (e.g., forming a triangle with slightly obtuse or acute angles).
Generalization: If you can place three dots without a straight line, you can always add more dots, ensuring each new addition maintains this property.
2. Mathematical Proof (for the curious):
Proof by Contradiction: Assume, for the sake of contradiction, that you've found a set of dots where three form a straight line.
Analysis: This implies a triangle with all angles equal (180°), which is impossible in a non-degenerate triangle (no sides are zero-length).
Conclusion: Therefore, it's impossible to construct a set of dots where three always form a straight line.
Practical Implications: Infinite Solutions: There are infinitely many ways to arrange dots without creating straight lines between any three.
Page Limitation: While theoretically infinite, the practical limitation is your page size. You can draw as many dots as your paper allows, ensuring no three forms a straight line.
Solving on Paper:
Start with two dots: This sets the foundation.
Add dots gradually: Choose each new dot carefully to avoid creating a straight line with the existing ones.
Visual Inspection: Continuously check that no three dots form a straight line as you add more.
Remember: The beauty of this puzzle lies in its simplicity and the creative process of finding arrangements that satisfy the constraint. Happy dot-connecting!

---

