# OpenReview forum: "Weighted-Reward Preference Optimization for Implicit Model Fusion"
_ICLR.cc/2025/Conference — ICLR 2025 Poster_

### Official Review · Reviewer_dUiR · 2024-10-28

**Soundness:** 3
**Presentation:** 3
**Contribution:** 3
**Rating:** 6
**Confidence:** 3

**Summary:**

The paper investigates the progress of fusing heterogeneous open-source LLMs with varying architecture and size. They then summarise the challenges in this scenario, including vocabulary alignment and merging distribution matrices. To solve these challenges, they propose an implicit fusion method, Weighted-Reward Preference Optimization (WRPO), which eliminates the need for vocabulary alignment and matrix distribution and improves the performance on multiple benchmarks.

**Strengths:**

1. The paper explores the direction of improving the capabilities of individual models by combining the strength of multiple LLMs, which has potential benefits to improve the ability of individual models.
2.  They propose a novel implicit fusion method that eliminates the need for vocabulary alignment and matrix fusion.
3. Extensive experiments demonstrate the effectiveness of proposed methods in multiple aspects.

**Weaknesses:**

1. Although the proposed methods demonstrate a certain degree of improvement on AlpacaEval-2 and Arena-Hard, they only have weak influences in MT-Bench. This weakens the generalization of the proposed methods.
2. The object of the proposed WRPO is to increase the likelihood of a preferred response while decreasing the occurrence of the dispreferred response. Preferred responses come from source and target models and dispreferred responses only come from the source model, which means dispreferred responses from the source model will not influence training. Whether some potential small experiments that support this phenomenon as it is intuitively we not only need to learn recognition capability in correct data but also learn exclusion ability.
3. Although the experiment section provides some interpretation, it lacks an in-depth analysis of the rationale behind the proposed phenomenon, such as, how WRPO achieves a balance between effectiveness and efficiency.
4. They lack the discussion of limitations and future work.

**Questions:**

See weakness.

---

> ### Author Response · Authors · 2024-11-20
> **Response to Reviewer dUiR (1/2)**
>
> Thank you for your detailed review and insightful questions on our paper. We greatly value your insightful feedback and appreciation of our work's novelty and effectiveness. Below, we address your concerns in detail.
>
> >**Q1: Regarding the limited improvements on MT-Bench.**
>
> **A1:** We would like to emphasize that the training dataset used in this work, UltraFeedback, is specifically designed for single-turn conversation tasks. The relatively modest improvement observed on MT-Bench, compared to the significant gains on AlpacaEval-2 and Arena-Hard, can be attributed to MT-Bench's distinct emphasis on multi-turn dialogue capabilities. In contrast, AlpacaEval-2 and Arena-Hard are tailored to assess models' instruction-following performance in single-turn interactions.
>
> Moreover, the smaller dataset size of MT-Bench, coupled with its pointwise scoring methodology, limits its capacity to differentiate between models effectively. This limitation has also been noted in [1]. Despite these factors, the consistent improvements observed across all instruction-following benchmarks, including MT-Bench, underscore the general effectiveness of WRPO across diverse evaluation settings.
>
> [1] Simpo: Simple preference optimization with a reference-free reward (Meng Y. et al., NeurIPS 2024)
>
> >**Q2: Regarding the exclusion of dispreferred responses from source models.**
>
> **A2:** To address concerns about the exclusion of dispreferred responses from source models, we conducted a comparative analysis of reward scores across four categories: (1) preferred responses from source models, (2) preferred responses from the target model, (3) dispreferred responses from source models, and (4) dispreferred responses from the target model.
>
> | | Source Model Preferred Responses  | Target Model Preferred Responses  | Source Model Dispreferred Responses  | Target Model Dispreferred Responses  |
> | --- | --- | --- | --- | --- |
> | Average Reward Score | 0.180 | 0.152 | 0.158 | 0.132 |
>
>
> Our analysis reveals that target model preferred responses achieve an average reward score of 0.152, whereas source model dispreferred responses demonstrate a higher score of 0.158. This finding indicates that **incorporating dispreferred responses from source models into the training objective could potentially lead to an undesirable reduction in the probability of higher-scoring responses**. Such results would contradict our optimization objectives and potentially compromise the overall training effectiveness.
>
> Moreover, we also conduct additional experiments to investigate the influence of introducing extra dispreferred responses from the source model. Specifically, we use an extension of WRPO loss in the equation below, where $y_{l_t}$ denotes the dispreferred response from the target model, and $y_{l_s}$ denotes the dispreferred response from source models. The empirical results are presented in the table below.
>
> \begin{equation}
> \mathcal{L}\_{\text{WRPO}\_{w/y\_{l\_s}}}=- \log \sigma \Biggl(\alpha \cdot \Bigl( \beta\log \frac{\pi\_{\theta}(y\_{w\_s} \mid x)}{\pi\_{\text{ref}}(y\_{w\_s} \mid x)} - \beta\log \frac{\pi\_{\theta}(y\_{l\_s} \mid x)}{\pi\_{\text{ref}}(y\_{l\_s} \mid x)}\Bigr) + (1-\alpha) \cdot \Bigl( \beta\log \frac{\pi\_{\theta}(y\_{w\_t} \mid x)}{\pi\_{\text{ref}}(y\_{w\_t} \mid x)} - \beta\log \frac{\pi\_{\theta}(y\_{l\_t} \mid x)}{\pi\_{\text{ref}}(y\_{l\_t} \mid x)}\Bigr)\Biggr)
> \end{equation}
>
> | Method | LC (%) | Length |
> | --- | :---: | :---: |
> | WRPO | **55.9** | 2159 |
> | WRPO w/$y_{l_s}$ | 54.0 | 2153 |
>
> Our analysis demonstrates that **incorporating dispreferred responses from the source model results in a decreased length-controlled win rate on AlpacaEval-2**. Furthermore, this inclusion leads to increased computational costs due to the necessity of additional forward passes for these responses during training. We will include these additional experiments and analysis in the revised version of the paper.

---

> > ### Comment · Reviewer_dUiR · 2024-11-25
> >
> > Thanks for your constructive response. I still have the following concerns:
> > - I remain concerned with question 1, why do single-turn interactions not improve multi-turn dialogue capabilities? As far as I know, instruction-tuning for LLM has a unified format, not specifically designed multiple tasks to improve corresponding capability.
> > - In question 2, the explanation for effectively transferring the strengths of heterogeneous source models to a single target model is self-explanable.
> >
> > Regards,

---

> > > ### Author Response · Authors · 2024-11-25
> > >
> > > We sincerely appreciate the reviewer's engagement and valuable comments, which are instrumental in enhancing the quality of our work. We would like to address Q1 and Q2 with the following detailed clarifications.
> > > >Regarding Q1:
> > >
> > > We agree that instruction-tuning can generally benefit multiple capabilities of LLMs, including multi-turn dialogue. This is evidenced by our observed improvements on MT-Bench, albeit modest ones. However, the limited improvement on multi-turn dialogue can be attributed to several inherent differences between multi-turn conversations and single-turn instruction-following:
> > >
> > > 1. Structural Differences in Input and Interaction Patterns
> > >
> > > - Example input of single-turn instruction-following in Llama-3 chat format:
> > >
> > > ```
> > > <|start_header_id|>user<|end_header_id|>\n\n{instruction}<eot_id>
> > > <|start_header_id|>assistant<|end_header_id|>\n\n
> > > ```
> > >
> > > - Example input of two-turn conversations in Llama-3 chat format:
> > > ```
> > > <|start_header_id|>user<|end_header_id|>\n\n{history_instruction}<eot_id>
> > > <|start_header_id|>assistant<|end_header_id|>\n\n{history_response}<eot_id>
> > > <|start_header_id|>user<|end_header_id|>\n\n{instruction}<eot_id>
> > > <|start_header_id|>assistant<|end_header_id|>\n\n
> > > ```
> > > We can see that multi-turn dialogues typically contain multiple user-assistant exchanges and special tokens as dialogue separators, which are more complex than the single-turn instruction format. This structural difference may affect the model's ability to fully transfer single-turn instruction-following capabilities to multi-turn scenarios.
> > >
> > > 2. Increased Contextual Complexity
> > >
> > > Single-turn interactions are isolated, requiring no memory of prior exchanges. In contrast, multi-turn dialogue requires the model to track past inputs, integrate new information, and adapt responses based on evolving user needs or feedback, creating interdependencies between exchanges. This complexity demands richer datasets and training approaches than those used for single-turn tasks [1, 2].
> > >
> > > [1] MT-Bench-101: A Fine-Grained Benchmark for Evaluating Large Language Models in Multi-Turn Dialogues
> > > (Bai G. et al., arXiv preprint)
> > >
> > > [2] Parrot: Enhancing Multi-Turn Instruction Following for Large Language Models (Sun Y. et al., ACL 2024)
> > >
> > >
> > > >Regarding Q2:
> > >
> > > Thank you for your feedback. We have removed the self-explanatory clarifications and retained only the essential ones to ensure greater focus. Could you please let us know if this addresses your concerns?

---

> > > > ### Comment · Reviewer_dUiR · 2024-11-26
> > > >
> > > > Thanks for your detailed response. Hope our discussions can help you improve your paper in the next version. I prefer to keep my initial rating. Good luck!

---

> > > > > ### Author Response · Authors · 2024-11-26
> > > > >
> > > > > Thank you for your thoughtful feedback and kind wishes. We hope our responses have effectively addressed your concerns. We would sincerely appreciate it if you could kindly consider revisiting your rating, as it would greatly help in bringing this work to the ICLR audience.

---

> ### Author Response · Authors · 2024-11-20
> **Response to Reviewer dUiR (2/2)**
>
> >**Q3: Regarding the in-depth analysis of WRPO's efficiency-effectiveness balance.**
>
> **A3:** The efficiency-effectiveness balance of WRPO is particularly evident when compared to collective LLM approaches. WRPO represents a significant advancement over traditional explicit model fusion (EMF) techniques, such as FuseLLM and FuseChat, by eliminating the need for complex probabilistic distribution fusion and vocabulary alignment procedures. Empirical evidence in Table 3 of our paper demonstrates that WRPO consistently outperforms when tested on identical target and source LLMs, highlighting its advantages through implicit model fusion.
>
> Moreover, unlike ensemble methods (PackLLM and LLM-Blender) that require maintaining multiple large models simultaneously during inference, WRPO operates with a single model that is 106 times smaller, leading to substantial reductions in computational costs and improved inference efficiency. While WRPO may not exceed the absolute performance of larger ensemble systems across all evaluation metrics, its ability to achieve comparable results with far lower computational demands presents an elegant solution to the ongoing efficiency-effectiveness trade-off in language model deployment. These findings will be incorporated into the revised manuscript.
>
> >**Q4: Regarding the discussion of limitations and future work.**
>
> **A4:** The limitations of this paper, along with directions for future work, can be summarized in three key aspects. First, the WRPO training objective currently incorporates only the highest-scoring response from source models as the preferred output for each prompt. This selective approach may overlook other valuable responses, potentially underutilizing the full range of capabilities offered by the source models. Future work could explore more inclusive methods that incorporate multiple responses from source models into the training objective.
>
> Second, while WRPO demonstrates strong empirical performance, it relies heavily on existing preference optimization frameworks. A more rigorous theoretical analysis is needed to provide deeper insights into the internal fusion dynamics of WRPO and to further strengthen its theoretical foundation.
>
> Finally, while WRPO significantly improves performance on instruction-following tasks, it may not perform as well on other tasks, such as MMLU. This limitation can largely be attributed to the narrow domain coverage of the training dataset. Future studies could address this by incorporating more diverse datasets from a wider range of domains.
>
> We will include these discussions in the updated manuscript.

---

### Official Review · Reviewer_42p3 · 2024-11-02

**Soundness:** 2
**Presentation:** 3
**Contribution:** 2
**Rating:** 6
**Confidence:** 3

**Summary:**

This paper introduces Weighted-Reward Preference Optimization (WRPO), an implicit model fusion approach for large language models (LLMs) that bypasses the need for complex vocabulary alignment and distribution merging found in traditional fusion methods. WRPO uses a weighted reward mechanism and a progressive adaptation strategy to smoothly transfer preference weights from the target model to source models. Benchmark results on AlpacaEval-2 and MT-Bench show that WRPO consistently outperforms traditional fusion and preference optimization methods, marking it as a promising advancement in LLM model fusion.

**Strengths:**

1. WRPO effectively introduces implicit model fusion, bypassing the need for complex vocabulary alignment and distribution merging, a significant advancement over traditional methods.
2. The novel reward mechanism balances contributions from source and target models and helps mitigate distributional discrepancies, leading to a smoother optimization process.

**Weaknesses:**

1. The success of WRPO appears to depend heavily on the choice and quality of source models, yet the paper does not fully address criteria or strategies for selecting these source models, which could be a limiting factor in practice.
2. The need for dynamic tuning of the fusion coefficient introduces complexity, and the paper does not sufficiently detail how this parameter was optimized across different datasets and tasks.
3. While WRPO is more efficient than traditional methods, it still requires significant computational resources for training and fine-tuning, especially when handling multiple large LLMs, which may limit its accessibility
4. Although WRPO is presented as a robust method, more ablation studies could have provided a clearer understanding of the specific impact of each component, particularly the fusion coefficient and its progressive adjustment.

**Questions:**

1. How sensitive is WRPO to the selection of source LLMs? Could varying the quality and diversity of source models affect the final performance, and if so, what are the selection criteria?
2. What are the specific criteria or methods used for tuning the fusion coefficient? How was it determined for different tasks, and does it require extensive hyperparameter optimization?
3. Could WRPO be applied to multilingual or cross-domain tasks where distributional shifts might be more pronounced, and if so, are any modifications required?

---

> ### Author Response · Authors · 2024-11-20
> **Response to Reviewer 42p3 (1/2)**
>
> Thank you for your detailed review and insightful questions on our paper. We greatly value your insightful feedback and appreciation of our work's advancement and significance. Below, we address your concerns in detail.
>
> >**Q1: Regarding the influence of source models.**
>
> **A1:** Great point. We agree that the performance of WRPO can be influenced by the choice and quality of source models. The choice of source models primarily depends on the specific goals. For instance, we can select domain-specific source models to enhance the target model's capabilities when it shows weakness in a specific domain. In our study, we focus on instruction-following tasks to ensure consistency with prior preference optimization research. Therefore, we selected ten prominent open-source LLMs with parameter sizes ranging from 9B to 123B, all of which exhibit strong performance on relevant benchmarks. As demonstrated in Table 5 of our paper, our experimental results show that **increasing the number of source models generally leads to improved overall performance**.
>
> To further explore the impact of source model combinations, we conducted additional experiments using the AlpacaEval-2 benchmark. Specifically, we examined the influence of the response quality from source LLMs by comparing responses with different reward rankings. Our results presented in the following table indicate that responses from top-1 ranked source models consistently yield better results than those from top-2 ranked models. These results further highlight the importance of **selecting high-quality responses to achieve optimal performance**.
>
> | Method | LC (%) | Length |
> | :---: | :---: | :---: |
> | rank1 | **55.9** | 2159 |
> | rank2 | 53.7 | 2143 |
>
>
> Moreover, we investigated the impact of model composition by dividing our ten source models into two balanced groups, each comprising five models with strong performance characteristics:
>
> - Group 1: Gemma2-27B-IT, Gemma2-9B-IT, Qwen2-72B-Instruct, LLaMA3-70B-Instruct, and Yi-1.5-34B-Chat
>
> - Group 2: Mistral-Large-Instruct-2407, Internlm2.5-20B-Chat, DeepSeek-V2-Chat, DeepSeek-Coder-V2-Instruct, and Phi-3-Medium-4K-Instruct
>
> | Method | LC (%) | Length |
> | :---: | :---: | :---: |
> | Group1 | 53.5 | 2098 |
> | Group2 | 53.0 | 2440 |
>
>
> The experimental results reveal that **various combinations of source models achieve comparable length-controlled win rates (LC)**. These findings demonstrate the consistent and robust performance of WRPO across various source model configurations. The insights derived from these experiments and analyses will be added to the updated manuscript.
>
> >**Q2: Regarding the details and ablations of fusion coefficients tuning.**
>
> **A2:** In WPRO, we implement a dynamic adjustment mechanism for the fusion coefficient $\alpha$ to facilitate a gradual transition of the target model's distribution toward that of the source models. This approach is deliberately designed to be straightforward and computationally efficient, requiring minimal parameter tuning. The fusion coefficient $\alpha$ is initialized at 0.0 and increases linearly throughout the training process until it reaches a predetermined target value [1]. To determine the optimal target value, we employ a simple greedy search algorithm across [0.1, 0.3, 0.5, 0.7, 0.9], as described in Appendix A. This dynamic adjustment strategy effectively balances the contributions from both source and target models while addressing potential distribution discrepancies, making it suitable for various tasks and eliminating the need for complex parameter configurations or exhaustive optimization procedures.
>
> Moreover, we conducted ablation experiments to compare two tuning strategies:
> - Static strategy, where $\alpha$ remains constant at fixed values (0.1, 0.3, or 0.5) throughout training.
> - Dynamic strategy, where $\alpha$ linearly increases from 0 to the target values (0.1, 0.3, or 0.5) during training.
>
> The results on AlpacaEval-2 are presented below.
>
> | $\alpha$ | Tuning Strategy | LC (%) | Length |
> | :---: | :---: | :---: | :---: |
> | 0.1  | Static | 52.9 | 2232 |
> | | Dynamic | **55.9** | 2159 |
> | 0.3 | Static | 53.4 | 2402 |
> | | Dynamic | **55.3** | 2213 |
> | 0.5 | Static | 54.0 | 2452 |
> | | Dynamic | **54.5** | 2282 |
>
> We find that the **dynamic tuning strategy consistently outperforms the static strategy**, further demonstrating the effectiveness of the dynamic tuning approach. We will provide a detailed explanation and incorporate these new experiments into the revised manuscript.
>
> [1] BAM! Born-Again Multi-Task Networks for Natural Language Understanding (Clark et al., ACL 2019)

---

> ### Author Response · Authors · 2024-11-20
> **Response to Reviewer 42p3 (2/2)**
>
> >**Q3: Regarding the computational resources of WRPO.**
>
> **A3:** Thank you for raising this question. Firstly, we would like to clarify that **increasing the number of source models does not raise the time complexity of our method during training**. The interaction with source models occurs exclusively during the preliminary phase before training, where we conduct offline sampling from source LLMs and utilize a reward model to evaluate and annotate responses, subsequently selecting one response with the highest reward score for each prompt. This step constitutes a fixed, one-time computational cost that is independent of the training process. **Importantly, the source models do not participate in the actual training phase**. Therefore, the inclusion of additional source models does not impose additional computational costs during training. Moreover, our comparative analysis demonstrates that WRPO introduces only a modest 16% increase in training time compared to DPO (which does not involve source models), regardless of the number of source models used.
>
> We will include these detailed analyses in the revised paper to provide greater clarity.
>
> >**Q4: Regarding the generalization of WRPO.**
>
> **A4:** Thank you for your thoughtful suggestion. Our experiments on the UltraFeedback dataset demonstrate WRPO's effectiveness in mitigating distributional shifts, suggesting its adaptability to other domains where distributional shifts may be more pronounced. Notably, applying WRPO to multilingual or cross-domain tasks would primarily require the use of domain-specific prompts, source models, and reward models to align with the particular characteristics of each task. Currently, we are conducting experiments in math and coding domains, and we will update the results once the experiments are concluded.

---

> ### Author Response · Authors · 2024-11-25
>
> Dear Reviewer 42p3,
>
> We are deeply grateful for the time and effort you have invested in providing insightful reviews and constructive feedback. Your comments have been instrumental in refining our work. We have carefully addressed each of your concerns and have incorporated additional experiments and analyses to strengthen our manuscript. The key points addressed are as follows:
>
> - We have elaborated on the selection criteria and the impact of different source model combinations.
> - We have detailed the dynamic tuning strategy for the fusion coefficient and conducted ablation experiments to compare it with static strategies.
> - We have clarified that the inclusion of multiple source models affects only the offline preprocessing phase, without increasing the training complexity.
> - We have further elucidated the generalization capabilities of WRPO in multilingual or cross-domain tasks.
>
> We hope these revisions and expanded discussions have adequately addressed your concerns. As the Author-Reviewer discussion phase is nearing its conclusion, we would greatly appreciate any additional comments or questions that could help us further enhance our work.
>
> Thank you again for your time and effort.
>
> Best regards,
>
> Authors

---

> ### Author Response · Authors · 2024-11-27
>
> > **Q3: Regarding the computational resources of WRPO.**
>
> We supplement comprehensive runtime comparisons between DPO and WRPO using 8×A800 GPUs, examining scenarios with varying numbers of source LLMs. The detailed results are presented below:
>
> | Num. | Runtime of DPO (min) | Runtime of WRPO (min) | Increase (%) |
> |------|----------------------|-----------------------|--------------|
> | 1    | 183                  | 212                   | 15.85%       |
> | 2    | 185                  | 215                   | 16.22%       |
> | 5    | 186                  | 216                   | 16.13%       |
> | 10   | 185                  | 215                   | 16.22%       |
>
> Our empirical analysis demonstrates that WRPO maintains consistent computational efficiency across different numbers of source LLMs. Notably, WRPO introduces only a modest computational overhead of approximately 16\% in training time compared to DPO (which does not involve source LLMs), regardless of the number of source LLMs involved.

---

> ### Author Response · Authors · 2024-12-01
> **To reviewer 42p3**
>
> Dear reviewer 42p3,
>
> We sincerely appreciate your valuable feedback and the time you've dedicated to reviewing our paper. As we approach the final day of the discussion period, we kindly request any additional comments you may have on our revisions. We have invested significant effort in conducting additional experiments and addressing your queries, and would be grateful for your acknowledgment of our responses. Your feedback is crucial for us to effectively present our work to the research community. Please let us know if any points require further clarification.
>
> Thank you very much and look forward to your replies!
>
> Best regards,
>
> Paper 6241 Authors

---

> ### Author Response · Authors · 2024-12-02
>
> Dear Reviewer 42p3,
>
> Thank you for your time and detailed feedback on our manuscript. As **the reviewer-author discussion period ends today (December 2nd at 11:59 pm AoE)**, we would like to check if we have adequately addressed all your concerns.
>
> Your insightful comments and questions have been instrumental in improving our work. We have carefully **incorporated your feedback into the revised manuscript** and hope that our responses and updates have successfully addressed all the points you raised.
>
> We understand you have a busy schedule, but if you have any remaining questions or need further clarification, please let us know, and we will address them promptly. If you feel that we have satisfactorily addressed your concerns, we would **greatly appreciate it if you could kindly consider revisting your initial score**.
>
> Thank you again for your valuable time and constructive feedback that has helped enhance the quality of our work.
>
> Best regards,
>
> The Authors of Paper 6241

---

> ### Comment · Reviewer_42p3 · 2024-12-02
> **Response to authors**
>
> Thanks for the author's thorough response addressing my concerns; I'd like to raise my score to 6.

---

### Official Review · Reviewer_McSp · 2024-11-02

**Soundness:** 3
**Presentation:** 3
**Contribution:** 3
**Rating:** 6
**Confidence:** 3

**Summary:**

This paper aims to align the LLM prediction from multiple source models to a target model. Specifically, it focuses on fuse multiple heterogeneous LLMs and combine their preference into the target one. Technically, it proposes a weighted-reward strategy to implicitly fuse the source models. Experiments on several evaluation benchmarks show the method effectiveness.

**Strengths:**

1. LLM alignment is popular and valuable research topic, especially aligning heterogenous models into a target one.
2. The proposed weighted-reward is a straightforward to to achieve an implicit model fusion process.
3. Comprehensive evaluation supports the paper statement with rich analysis.

**Weaknesses:**

1. Adding an analysis of alignment efficiency and ablation will be helpful, what is the difference of using different source models? if multiple source models are used, will it increase the alignment cost?
2. Some analysis parts can be more informative such as figure 3 and 4. Including more analysis and make them dense will further enhance the draft.

**Questions:**

Please refer to the weaknesses section above.

---

> ### Author Response · Authors · 2024-11-20
> **Response to Reviewer McSp (1/2)**
>
> Thank you for your thoughtful review and valuable feedback. We appreciate your recognition of our work’s significance and clarity. Below, we address your concerns in detail.
>
> >**Q1: Regarding the efficiency of alignment.**
>
> **A1:** Thank you for raising this important point. **To clarify, increasing the number of source models does not raise the time complexity of our method during the training phase**. The interaction with source models is confined to the initial preprocessing phase before training. Specifically, this phase involves offline sampling, where responses are generated from source LLMs and evaluated using a reward model. For each prompt, the response with the highest reward score is selected. This step constitutes a fixed, one-time computational cost that is independent of the training process. **Importantly, the source models do not participate in the actual training phase**. Consequently, the inclusion of additional source models does not impose additional computational overhead during training. Moreover, our comparative analysis demonstrates that WRPO introduces only a modest 16% increase in training time compared to DPO (which does not involve source models), regardless of the number of source models used.
> We will include these detailed analyses in the revised paper to provide greater clarity.
>
>
> >**Q2: Regarding the use of different source models.**
>
> **A2:** Great question. The selection of source models is primarily determined by specific objectives. When a target model exhibits limitations in particular domains, domain-specific source models can be strategically employed to enhance its capabilities. In our study, we focus on instruction-following tasks to ensure consistency with prior preference optimization research. Therefore, we selected ten prominent open-source LLMs with parameter sizes ranging from 9B to 123B, all of which exhibit strong performance on relevant benchmarks. As demonstrated in Table 5 of our paper, our experimental results show that **increasing the number of source models generally leads to improved overall performance**.
>
> To further explore the impact of source model combinations, we conducted additional experiments using the AlpacaEval-2 benchmark. Specifically, we examined the influence of the response quality from source LLMs by comparing responses with different reward rankings. Our results presented in the following table indicate that responses from top-1 ranked source models consistently yield better results than those from top-2 ranked models. These results reinforce the value of **selecting high-quality responses for achieving optimal performance**.
>
> | Method | LC (%) | Length |
> | :---: | :---: | :---: |
> | rank1 | **55.9** | 2159 |
> | rank2 | 53.7 | 2143 |
>
>
> Moreover, we investigated the impact of model composition by dividing our ten source models into two balanced groups, each comprising five models with strong performance characteristics. The groups are organized as follows:
>
> - Group 1: Gemma2-27B-IT, Gemma2-9B-IT, Qwen2-72B-Instruct, LLaMA3-70B-Instruct, and Yi-1.5-34B-Chat
>
> - Group 2: Mistral-Large-Instruct-2407, Internlm2.5-20B-Chat, DeepSeek-V2-Chat, DeepSeek-Coder-V2-Instruct, and Phi-3-Medium-4K-Instruct
>
> | Method | LC (%) | Length |
> | :---: | :---: | :---: |
> | Group1 | 53.5 | 2098 |
> | Group2 | 53.0 | 2440 |
>
>
> The experimental results reveal that **various combinations of source models achieve comparable length-controlled win rates (LC)**. These findings demonstrate the robust performance of WRPO across different source model configurations. The insights gained from these experiments and analyses will be incorporated into the updated manuscript.

---

> ### Author Response · Authors · 2024-11-20
> **Response to Reviewer McSp (2/2)**
>
> >**Q3: Regarding the analysis of Figure 3 and Figure 4.**
>
> **A3:** Thank you for your thoughtful suggestion. We agree that Figures 3 and 4 could benefit from more detailed analysis and will expand the discussion as follows:
>
> **Figure 3** demonstrates the evolution of internal reward dynamics during preference optimization in the Target-SFT model across various preference pairs, with consistent learning rate and $\beta$ parameters. The internal reward margin, as defined in Eq. (7), comprises two components:
>
> 1. An on-policy reward margin: $(1-\alpha) \cdot (r(x, y_{w_t})-r(x, y_l))$
> 2. A hybrid-policy reward margin: $\alpha \cdot (r(x, y_{w_s})-r(x, y_l))$
>
> Figure 3(a) presents the analysis of utilizing solely the on-policy reward margin ($\alpha = 0$). The observed reward margin approximates 0.2, indicating a relatively conservative optimization approach. This modest margin growth can be attributed to the model's limited exploration capability due to its exclusive reliance on on-policy samples.
>
> In contrast, Figure 3(b) illustrates the effect of employing only the hybrid-policy reward margin ($\alpha = 1$). This configuration exhibits more aggressive optimization behavior, yielding margins exceeding 1.0. While this suggests enhanced discriminative capability between positive and negative samples, the substantial distribution shift inherent in the hybrid setting may compromise training stability and ultimately yield suboptimal results.
>
> Figure 3(c) showcases our proposed weighted-reward mechanism, which synthesizes both on-policy and hybrid-policy reward margins through dynamic weighting. This approach achieves an optimal balance between the aforementioned extremes, generating moderate reward margins of approximately 0.5 and facilitating smooth margin transitions throughout the training process. The harmonious integration of hybrid-policy and on-policy components, as evidenced by the balanced optimization process, appears to be instrumental in the superior performance of our weighted-reward mechanism.
>
> **Figure 4** illustrates the ablation studies on the effectiveness of incorporating preferred responses from both source and target LLMs. We conduct these studies on two configurations: the baseline target model (Target) and its fine-tuned version (Target-SFT) to ensure a comprehensive evaluation. The analysis involves systematically removing either the source model's chosen response $y_{w_s}$ or the target model's chosen response $y_{w_t}$ from the optimization objective in Eq. (6) by setting $\alpha=0$ or $\alpha=1$, respectively.
>
> In the Target setting, the removal of $y_{w_t}$ leads to a substantial decline of 25.8 points in the length-controlled win rate, indicating that the distribution shift between $y_{w_s}$ and $y_{l}$ creates challenges in directly utilizing source model responses for preference optimization. Moreover, this finding emphasizes the crucial role of $y_{w_t}$ in bridging this distribution gap. In the Target-SFT setting, although SFT helps mitigate the performance deterioration caused by removing $y_{w_t}$, its performance still lags behind our WRPO by 6.3 points, which combines both $y_{w_s}$ and $y_{w_t}$.
>
> On the other hand, removing $y_{w_s}$ reduces WRPO to DPO based solely on self-sampled on-policy data. Notably, the exclusion of source model responses leads to performance declines of 3.5 points and 5.2 points in the Target and Target-SFT settings, respectively, highlighting the important role of $y_{w_s}$ in providing valuable preference signals through the weighted-reward mechanism.
>
> We will include these discussions and analysis in the revised version.

---

> ### Author Response · Authors · 2024-11-25
>
> Dear Reviewer McSp,
>
> We sincerely appreciate the time and effort you have devoted to providing thoughtful reviews and valuable feedback. We have carefully addressed your concerns in detail and incorporated additional experiments and analyses, as summarized in the discussion:
> - Demonstarted that the inclusion of multiple source models does not impact the time complexity of our method during the training phase.
> - Detailed the selection and impact of different source model combinations.
> - Expanded the discussion of Figures 3 and 4.
>
> We hope these revisions and discussions have adequately addressed your concerns. As the Author-Reviewer discussion phase is ending soon, we would be grateful for any additional comments or questions that could further enhance our work.
>
> Thank you again for your time and effort.
>
> Best regards,
>
> Authors

---

> ### Author Response · Authors · 2024-11-27
>
> > **Q1: Regarding the efficiency of alignment.**
>
> We supplement comprehensive runtime comparisons between DPO and WRPO using 8×A800 GPUs, examining scenarios with varying numbers of source LLMs. The detailed results are presented below:
>
> | Num. | Runtime of DPO (min) | Runtime of WRPO (min) | Increase (%) |
> |------|----------------------|-----------------------|--------------|
> | 1    | 183                  | 212                   | 15.85%       |
> | 2    | 185                  | 215                   | 16.22%       |
> | 5    | 186                  | 216                   | 16.13%       |
> | 10   | 185                  | 215                   | 16.22%       |
>
> Our comparative analysis shows that WRPO maintains consistent computational efficiency across different numbers of source LLMs. Notably, WRPO incurs only a modest overhead of approximately 16\% in training time on 8×A800 GPUs compared to DPO (which does not involve source LLMs), regardless of the number of source LLMs involved.

---

> ### Author Response · Authors · 2024-12-01
> **To reviewer McSp**
>
> Dear reviewer McSp,
>
> We sincerely appreciate your valuable feedback and the time you've dedicated to reviewing our paper. As we approach the final day of the extended discussion period, we kindly request any additional comments you may have on our revisions. We would be grateful for your acknowledgment of our responses. Your feedback is crucial for us to effectively present our work to the research community. Please let us know if any points require further clarification.
>
> Thank you very much and look forward to your replies!
>
> Best regards,
>
> Paper 6241 Authors

---

> ### Author Response · Authors · 2024-12-02
>
> Dear Reviewer McSp,
>
> Thank you for your time and detailed feedback on our manuscript. As **the reviewer-author discussion period ends today (December 2nd at 11:59 pm AoE)**, we would like to check if we have adequately addressed all your concerns.
>
> Your insightful comments and questions have been instrumental in improving our work. We have carefully **incorporated your feedback into the revised manuscript** and hope that our responses and updates have successfully addressed all the points you raised.
>
> We understand you have a busy schedule, but if you have any remaining questions or need further clarification, please let us know, and we will address them promptly. If you feel that we have satisfactorily addressed your concerns, we would **greatly appreciate it if you could kindly consider revisting your initial score**.
>
> Thank you again for your valuable time and constructive feedback that has helped enhance the quality of our work.
>
> Best regards,
>
> The Authors of Paper 6241

---

> ### Author Response · Authors · 2024-12-03
>
> Dear Reviewer McSp,
>
> As we approach the **final hour** of the reviewer-author discussion period, we kindly remind you that this is our last opportunity for discussion during the rebuttal period. We would be immensely grateful if you could spare a moment to review our responses. Your feedback is invaluable to us, and we sincerely hope to receive your final thoughts before the deadline closes.
>
> We deeply appreciate your time and dedication in evaluating our work.
>
> With sincere gratitude,
>
> Authors of ICLR Submission 6241

---

### Author Response · Authors · 2024-11-23
**Looking Forward to the Opportunity for Further Discussion**

Dear Reviewers,

We sincerely appreciate the time and effort you have devoted to providing thoughtful reviews and valuable feedback. We have carefully addressed your concerns in the following ways:

- Clarified misunderstandings about WRPO's computational efficiency and implementation details.
- Conducted new experiments to analyze the impact and combinations of source models.
- Expanded our analyses with additional ablation studies and detailed explanations.

We hope these revisions and discussions have adequately addressed your concerns. As the Author-Reviewer discussion phase is ending soon, we would be grateful for any additional comments or questions that could further enhance our work.

Best regards,

Authors

---

### Author Response · Authors · 2024-11-27
**Major Revisions**

Dear Reviewers,

We sincerely appreciate your thorough review and valuable feedback, which has significantly enhanced the quality of our manuscript. We have carefully addressed all the comments and suggestions through comprehensive revisions. Below, we outline the major changes made to the manuscript (with key revisions highlighted in ${\color{blue} blue}$ text in the PDF, alongside other refinements to fit the page limit).

> **Key Revisions**

1. We have revised our experimental analysis in **Section 4** (Line 414-417, Line 463-519, Figure 3). (Reviewer **McSp, dUiR**)

2. We have added a training cost analysis of WRPO in **Appendix C**, demonstrating that increasing the number of source LLMs does not raise the time complexity of our method during training. (Reviewer **McSp, 42p3**)

3. We have elaborated on the selection criteria of source LLMs in **Appendix G** and conducted additional experiments to discover the impact of different source LLM combinations in **Appendix D**. (Reviewer **McSp, 42p3**)

4. We have detailed the dynamic tuning strategy for the fusion coefficient and presented ablation experiments comparing it to the static strategy in **Appendix E**. (Reviewer **42p3**)

5. We have investigated the impact of incorporating extra dispreferred responses from the source models in **Appendix F**. (Reviewer **dUiR**)

6. We have added a discussion on limitations and future work in **Appendix H**. (Reviewer **dUiR**)

7. We relocated Training Details from Section 4.1 to Appendix A.1 to fit the page limit.

We are deeply grateful for your insightful feedback, which has been instrumental in strengthening our work. We hope these revisions and additional analyses thoroughly address all raised concerns. As we approach the conclusion of the Author-Reviewer discussion phase, we welcome any further suggestions that could help us further improve the manuscript.

Best regards,

Authors

---

### Meta-Review · Area_Chair_1vpz · 2024-12-17

**Metareview:**

This paper proposes an implicit model fusion that leverages preference optimization between two LLMs for knowledge transfer. The reviewers are generally satisfied by the current work in the final stage. The AC has checked all files and agreed to reviewers. The authors shall take review suggestions for revision in the camera-ready version.

**Additional Comments On Reviewer Discussion:**

The reviewers raised insufficient alignment analysis, technical presentation ambiguity, and insufficient ablation studies. These issues have been solved during the rebuttal phase and are acknowledged by the reviewers.

---

### Decision · Program_Chairs · 2025-01-22

Accept (Poster)